# Functional Variational Bayesian Neural Networks

**Shengyang Sun**[*†]**, Guodong Zhang**[*†]**, Jiaxin Shi**[*‡]**, Roger Grosse**[†]

[†]University of Toronto, [†]Vector Institute, [‡]Tsinghua University

{ssy, gdzhang, rgrosse}@cs.toronto.edu, shijx15@mails.tsinghua.edu.cn

## Abstract

Variational Bayesian neural networks (BNNs) perform variational inference over weights, but it is difficult to specify meaningful priors and approximate posteriors in a high-dimensional weight space. We introduce functional variational Bayesian neural networks (fBNNs), which maximize an Evidence Lower BOund (ELBO) defined directly on stochastic processes, i.e. distributions over functions. We prove that the KL divergence between stochastic processes equals the supremum of marginal KL divergences over all finite sets of inputs. Based on this, we introduce a practical training objective which approximates the functional ELBO using finite measurement sets and the spectral Stein gradient estimator. With fBNNs, we can specify priors entailing rich structures, including Gaussian processes and implicit stochastic processes. Empirically, we find fBNNs extrapolate well using various structured priors, provide reliable uncertainty estimates, and scale to large datasets.

## 1 Introduction

Bayesian neural networks (BNNs) (Hinton & Van Camp, 1993; Neal, 1995) have the potential to combine the scalability, flexibility, and predictive performance of neural networks with principled Bayesian uncertainty modelling. However, the practical effectiveness of BNNs is limited by our ability to specify meaningful prior distributions and by the intractability of posterior inference. Choosing a meaningful prior distribution over network weights is difficult because the weights have a complicated relationship to the function computed by the network. Stochastic variational inference is appealing because the update rules resemble ordinary backprop (Graves, 2011; Blundell et al., 2015), but fitting accurate posterior distributions is difficult due to strong and complicated posterior dependencies (Louizos & Welling, 2016; Zhang et al., 2018; Shi et al., 2018a).

In a classic result, Neal (1995) showed that under certain assumptions, as the width of a shallow BNN was increased, the limiting distribution is a Gaussian process (GP). Lee et al. (2018) recently extended this result to deep BNNs. Deep Gaussian Processes (DGP) (Cutajar et al., 2017; Salimbeni & Deisenroth, 2017) have close connections to BNNs due to similar deep structures. However, the relationship of finite BNNs to GPs is unclear, and practical variational BNN approximations fail to match the predictions of the corresponding GP. Furthermore, because the previous analyses related specific BNN architectures to specific GP kernels, it's not clear how to design BNN architectures for a given kernel. Given the rich variety of structural assumptions that GP kernels can represent (Rasmussen & Williams, 2006; Lloyd et al., 2014; Sun et al., 2018), there remains a significant gap in expressive power between BNNs and GPs (not to mention stochastic processes more broadly).

In this paper, we perform variational inference directly on the distribution of functions. Specifically, we introduce functional variational BNNs (fBNNs), where a BNN is trained to produce a distribution of functions with small KL divergence to the true posterior over functions. We prove that the KL divergence between stochastic processes can be expressed as the supremum of marginal KL divergences at finite sets of points. Based on this, we present functional ELBO (fELBO) training objective. Then we introduce a GAN-like minimax formulation and a sampling-based approximation for functional variational inference. To approximate the marginal KL divergence gradients, we adopt the recently proposed spectral Stein gradient estimator (SSGE) (Shi et al., 2018b).

---

[*]Equal contribution.

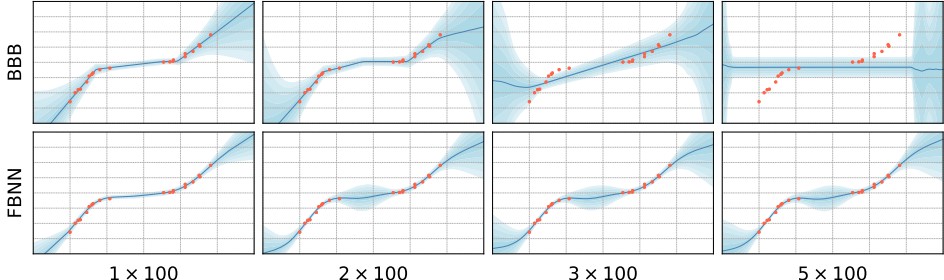

**Figure 1:** Predictions on the toy funcction $y = x^3$. Here $a \times b$ represents $a$ hidden layers of $b$ units. Red dots are 20 training points. The blue curve is the mean of final prediction, and the shaded areas represent standard derivations. We compare fBNNs and Bayes-by-Backprop (BBB). For BBB, which performs weight-space inference, varying the network size leads to drastically different predictions. For fBNNs, which perform function-space inference, we observe consistent predictions for the larger networks. Note that the $1 \times 100$ factorized Gaussian fBNNs network is not expressive enough to generate diverse predictions.

Our fBNNs make it possible to specify stochastic process priors which encode richly structured dependencies between function values. This includes stochastic processes with explicit densities, such as GPs which can model various structures like smoothness and periodicity (Lloyd et al., 2014; Sun et al., 2018). We can also use stochastic processes with implicit densities, such as distributions over piecewise linear or piecewise constant functions. Furthermore, in contrast with GPs, fBNNs efficiently yield explicit posterior samples of the function. This enables fBNNs to be used in settings that require explicit minimization of sampled functions, such as Thompson sampling (Thompson, 1933; Russo & Van Roy, 2016) or predictive entropy search (Hernández-Lobato et al., 2014; Wang & Jegelka, 2017).

One desideratum of Bayesian models is that they behave gracefully as their capacity is increased (Rasmussen & Ghahramani, 2001). Unfortunately, ordinary BNNs don't meet this basic requirement: unless the asymptotic regime is chosen very carefully (e.g. Neal (1995)), BNN priors may have undesirable behaviors as more units or layers are added. Furthermore, larger BNNs entail more difficult posterior inference and larger description length for the posterior, causing degeneracy for large networks, as shown in Figure 1. In contrast, the prior of fBNNs is defined directly over the space of functions, thus the BNN can be made arbitrarily large without changing the functional variational inference problem. Hence, the predictions behave well as the capacity increases.

Empirically, we demonstrate that fBNNs generate sensible extrapolations for both explicit periodic priors and implicit piecewise priors. We show fBNNs outperform competing approaches on both small scale and large scale regression datasets. fBNNs' reliable uncertainty estimates enable state-of-art performance on the contextual bandits benchmark of Riquelme et al. (2018).

## 2 BACKGROUND

### 2.1 VARIATIONAL INFERENCE FOR BAYESIAN NEURAL NETWORKS

Given a dataset $\mathcal{D} = \{(\mathbf{x}_i, y_i)\}_{i=1}^{n}$, a Bayesian neural network (BNN) is defined in terms of a prior $p(\mathbf{w})$ on the weights, as well as the likelihood $p(\mathcal{D}|\mathbf{w})$. Variational Bayesian methods (Hinton & Van Camp, 1993; Graves, 2011; Blundell et al., 2015) attempt to fit an approximate posterior $q(\mathbf{w})$ to maximize the evidence lower bound (ELBO):

$$\mathcal{L}_q = \mathbb{E}_q[\log p(\mathcal{D}|\mathbf{w})] - \mathrm{KL}[q(\mathbf{w})\|p(\mathbf{w})]. \tag{1}$$

The most commonly used variational BNN training method is Bayes By Backprop (BBB) (Blundell et al., 2015), which uses a fully factorized Gaussian approximation to the posterior, i.e. $q(\mathbf{w}) = \mathcal{N}(\mathbf{w}; \boldsymbol{\mu}, \mathrm{diag}(\boldsymbol{\sigma}^2))$. Using the reparameterization trick (Kingma & Welling, 2013), the gradients of ELBO towards $\mu, \sigma$ can be computed by backpropagation, and then be used for updates.

Most commonly, the prior $p(\mathbf{w})$ is chosen for computational convenience; for instance, independent Gaussian or Gaussian mixture distributions. Other priors, including log-uniform priors (Kingma et al., 2015; Louizos et al., 2017) and horseshoe priors (Ghosh et al., 2018; Louizos et al., 2017), were proposed for specific purposes such as model compression and model selection. But the relationships of weight-space priors to the functions computed by networks are difficult to characterize.

## 2.2 Stochastic Processes

A stochastic process (Lamperti, 2012) $F$ is typically defined as a collection of random variables, on a probability space $(\Omega, \mathcal{F}, P)$. The random variables, indexed by some set $\mathcal{X}$, all take values in the same mathematical space $\mathcal{Y}$. In other words, given a probability space $(\Omega, \Sigma, P)$, a stochastic process can be simply written as $\{F(\mathbf{x}) : \mathbf{x} \in \mathcal{X}\}$. For any point $\omega \in \Omega$, $F(\cdot, \omega)$ is a sample function mapping index space $\mathcal{X}$ to space $\mathcal{Y}$, which we denote as $f$ for notational simplicity.

For any finite index set $\mathbf{x}_{1:n} = \{\mathbf{x}_1, ..., \mathbf{x}_n\}$, we can define the finite-dimensional marginal joint distribution over function values $\{F(\mathbf{x}_1), \cdots, F(\mathbf{x}_n)\}$. For example, Gaussian Processes have marginal distributions as multivariate Gaussians.

The Kolmogorov Extension Theorem (Øksendal, 2003) shows that a stochastic process can be characterized by marginals over all finite index sets. Specifically, for a collection of joint distributions $\rho_{\mathbf{x}_{1:n}}$, we can define a stochastic process $F$ such that for all $\mathbf{x}_{1:n}$, $\rho_{\mathbf{x}_{1:n}}$ is the marginal joint distribution of $F$ at $\mathbf{x}_{1:n}$, as long as $\rho$ satisfies the following two conditions:

**Exchangeability.** For any permutation $\pi$ of $\{1, \cdots, n\}$, $\rho_{\pi(\mathbf{x}_{1:n})}(\pi(y_{1:n})) = \rho_{\mathbf{x}_{1:n}}(y_{1:n})$.

**Consistency.** For any $1 \leq m \leq n$, $\rho_{\mathbf{x}_{1:m}}(y_{1:m}) = \int \rho_{\mathbf{x}_{1:n}}(y_{1:n}) dy_{m+1:n}$.

## 2.3 Spectral Stein Gradient Estimator (SSGE)

When applying Bayesian methods to modern probabilistic models, especially those with neural networks as components (e.g., BNNs and deep generative models), it is often the case that we have to deal with intractable densities. Examples include the marginal distribution of a non-conjugate model (e.g., the output distribution of a BNN), and neural samplers such as GANs (Goodfellow et al., 2014). A shared property of these distributions is that they are defined through a tractable sampling process, despite the intractable density. Such distributions are called implicit distributions (Huszár, 2017).

The Spectral Stein Gradient Estimator (SSGE) (Shi et al., 2018b) is a recently proposed method for estimating the log density derivative function of an implicit distribution, only requiring samples from the distribution. Specifically, given a continuous differentiable density $q(\mathbf{x})$, and a positive definite kernel $k(\mathbf{x}, \mathbf{x}')$ in the Stein class (Liu et al., 2016) of $q$, they show

$$\nabla_{x_i} \log q(\mathbf{x}) = -\sum_{j=1}^{\infty} \left[ \mathbb{E}_q \nabla_{x_i} \psi_j(\mathbf{x}) \right] \psi_j(\mathbf{x}), \tag{2}$$

where $\{\psi_j\}_{j \geq 1}$ is a series of eigenfunctions of $k$ given by Mercer's theorem: $k(\mathbf{x}, \mathbf{x}') = \sum_j \mu_j \psi_j(\mathbf{x}) \psi_j(\mathbf{x}')$. The Nyström method (Baker, 1997; Williams & Seeger, 2001) is used to approximate the eigenfunctions $\psi_j(\mathbf{x})$ and their derivatives. The final estimator is given by truncating the sum in Equation (2) and replacing the expectation by Monte Carlo estimates.

# 3 Functional Variational Bayesian Neural Networks

## 3.1 Functional Evidence Lower Bound (fELBO)

We introduce function space variational inference analogously to weight space variational inference (see Section 2.1), except that the distributions are over functions rather than weights. We assume a stochastic process prior $p$ over functions $f : \mathcal{X} \to \mathcal{Y}$. This could be a GP, but we also allow stochastic processes without closed-form marginal densities, such as distributions over piecewise linear functions. For the variational posterior $q_\phi \in \mathcal{Q}$, we consider a neural network architecture with stochastic weights and/or stochastic inputs. Specifically, we sample a function from $q$ by sampling a random noise vector $\xi$ and defining $f(\mathbf{x}) = g_\phi(\mathbf{x}, \xi)$ for some function $g_\phi$. For example, standard weight space BNNs with factorial Gaussian posteriors can be viewed this way using the reparameterization trick (Kingma & Welling, 2013; Blundell et al., 2015). (In this case, $\phi$ corresponds to the means and variances of all the weights.) Note that because a *single* vector $\xi$ is shared among all input locations, it corresponds to randomness in the *function*, rather than observation noise; hence, the sampling of $\xi$ corresponds to epistemic, rather than aleatoric, uncertainty (Depeweg et al., 2017).

Functional variational inference maximizes the functional ELBO (fELBO), akin to the weight space ELBO in Equation (1), except that the distributions are over functions rather than weights.

$$\mathcal{L}(q) := \mathbb{E}_q[\log p(\mathcal{D}|f)] - \text{KL}[q||p]. \tag{3}$$

Here $\text{KL}[q\|p]$ is the KL divergence between two stochastic processes. As pointed out in Matthews et al. (2016), it does not have a convenient form as $\int \log \frac{q(f)}{p(f)} q(f) df$ due to there is no infinite-dimensional Lebesgue measure (Eldredge, 2016). Since the KL divergence between stochastic processes is difficult to work with, we reduce it to a more familiar object: KL divergence between the marginal distributions of function values at finite sets of points, which we term *measurement sets*. Specifically, let $\mathbf{X} \in \mathcal{X}^n$ denote a finite measurement set and $P_{\mathbf{X}}$ the marginal distribution of function values at $\mathbf{X}$. We equate the function space KL divergence to the supremum of marginal KL divergences over all finite measurement sets:

**Theorem 1.** *For two stochastic processes $P$ and $Q$,*

$$\text{KL}[P\|Q] = \sup_{n\in\mathbb{N},\mathbf{X}\in\mathcal{X}^n} \text{KL}[P_{\mathbf{X}}\|Q_{\mathbf{X}}]. \tag{4}$$

Roughly speaking, this result follows because the $\sigma$-algebra constructed with the Kolmogorov Extension Theorem (Section 2.2) is generated by *cylinder sets* which depend only on finite sets of points. A full proof is given in Appendix A.

**fELBO.** Using this characterization of the functional KL divergence, we rewrite the fELBO:

$$\begin{aligned}
\mathcal{L}(q) &= \mathbb{E}_q[\log p(\mathcal{D}|f)] - \sup_{n\in\mathbb{N},\mathbf{X}\in\mathcal{X}^n} \text{KL}[q(\mathbf{f}^{\mathbf{X}})||p(\mathbf{f}^{\mathbf{X}})] \\
&= \inf_{n\in\mathbb{N},\mathbf{X}\in\mathcal{X}^n} \sum_{(\mathbf{x}_i,y_i)\in\mathcal{D}} \mathbb{E}_q[\log p(y_i|f(\mathbf{x}_i))] - \text{KL}[q(\mathbf{f}^{\mathbf{X}})||p(\mathbf{f}^{\mathbf{X}})] \\
&:= \inf_{n\in\mathbb{N},\mathbf{X}\in\mathcal{X}^n} \mathcal{L}_{\mathbf{X}}(q).
\end{aligned} \tag{5}$$

We also denote $\mathcal{L}_n(q) := \inf_{\mathbf{X}\in\mathcal{X}^n} \mathcal{L}_{\mathbf{X}}(q)$ for the restriction to sets of $n$ points. This casts maximizing the fELBO as a two-player zero-sum game analogous to a generative adversarial network (GAN) (Goodfellow et al., 2014): one player chooses the stochastic network, and the adversary chooses the measurement set. Note that the infimum may not be attainable, because the size of the measurement sets is unbounded. In fact, the function space KL divergence may be infinite, for instance if the prior assigns measure zero to the set of functions representable by a neural network (Arjovsky & Bottou, 2017). Observe that GANs face the same issue: because a generator network is typically limited to a submanifold of the input domain, an ideal discriminator could discriminate real and fake images perfectly. However, by limiting the capacity of the discriminator, one obtains a useful training objective. By analogy, we obtain a well-defined and practically useful training objective by restricting the measurement sets to a fixed finite size. This is discussed further in the next section.

## 3.2 Choosing the Measurement Set

As discussed above, we approximate the fELBO using finite measurement sets to have a well-defined and practical optimization objective. We now discuss how to choose the measurement sets.

**Adversarial Measurement Sets.** The minimax formulation of the fELBO naturally suggests a two-player zero-sum game, analogous to GANs, whereby one player chooses the stochastic network representing the posterior, and the adversary chooses the measurement set.

$$\max_{q\in\mathcal{Q}} \mathcal{L}_m(q) := \max_{q\in\mathcal{Q}} \min_{\mathbf{X}\in\mathcal{X}^m} \mathcal{L}_{\mathbf{X}}(q). \tag{6}$$

We adopt concurrent optimization akin to GANs (Goodfellow et al., 2014). In the inner loop, we minimize $\mathcal{L}_{\mathbf{X}}(q)$ with respect to $\mathbf{X}$; in the outer loop, we maximize $\mathcal{L}_{\mathbf{X}}(q)$ with respect to $q$.

Unfortunately, this approach did not perform well in terms of generalization. The measurement set which maximizes the KL term is likely to be close to the training data, since these are the points where one has the most information about the function. But the KL term is the only part of the fELBO encouraging the network to match the prior structure. Hence, if the measurement set is close to the training data, then nothing will encourage the network to exploit the structured prior for extrapolation.

**Sampling-Based Measurement Sets.** Instead, we adopt a sampling-based approach. In order to use a structured prior for extrapolation, the network needs to match the prior structure both near the training data and in the regions where it must make predictions. Therefore, we sample measurement sets which include both (a) random training inputs, and (b) random points from the domain where one is interested in making predictions. We replace the minimization in Equation (6) with a sampling distribution $c$, and then maximize the expected $\mathcal{L}_{\mathbf{X}}(q)$:

$$\max_{q \in \mathcal{Q}} \mathbb{E}_{\mathcal{D}_s} \mathbb{E}_{\mathbf{X}^M \sim c} \mathcal{L}_{\mathbf{X}^M, \mathbf{X}^{D_s}}(q). \tag{7}$$

where $\mathbf{X}^M$ are $M$ points independently drawn from $c$.

**Consistency.** With the restriction to finite measurement sets, one only has an upper bound on the true fELBO. Unfortunately, this means the approximation is not a lower bound on the log marginal likelihood (log-ML) $\log p(\mathcal{D})$. Interestingly, if the measurement set is chosen to include all of the training inputs, then $\mathcal{L}(q)$ is in fact a log-ML lower bound:

**Theorem 2** (Lower Bound). *If the measurement set $\mathbf{X}$ contains all the training inputs $\mathbf{X}^D$, then*

$$\mathcal{L}_{\mathbf{X}}(q) = \log p(\mathcal{D}) - \mathrm{KL}[q(\mathbf{f}^{\mathbf{X}}) \| p(\mathbf{f}^{\mathbf{X}} | \mathcal{D})] \leq \log p(\mathcal{D}). \tag{8}$$

The proof is given in Appendix B.1.

To better understand the relationship between adversarial and sampling-based inference, we consider the idealized scenario where the measurement points in both approaches include all training locations, i.e., $\mathbf{X} = \{\mathbf{X}^D, \mathbf{X}^M\}$. Let $\mathbf{f}^M, \mathbf{f}^D$ be the function values at $\mathbf{X}^M, \mathbf{X}^D$, respectively. By Theorem 2,

$$\mathcal{L}_{\mathbf{X}^M, \mathbf{X}^D}(q) = \log p(\mathcal{D}) - \mathrm{KL}[q(\mathbf{f}^M, \mathbf{f}^D) \| p(\mathbf{f}^M, \mathbf{f}^D | \mathcal{D})]. \tag{9}$$

So maximizing $\mathcal{L}_{\mathbf{X}^M, \mathbf{X}^D}(q)$ is equivalent to minimizing the KL divergence from the true posterior on points $\mathbf{X}^M, \mathbf{X}^D$. Based on this, we have the following consistency theorem that helps justify the use of adversarial and sampling-based objectives with finite measurement points.

**Corollary 3** (Consistency under finite measurement points). *Assume that the true posterior $p(f|\mathcal{D})$ is a Gaussian process and the variational family $\mathcal{Q}$ is all Gaussian processes. We have the following results if $M > 1$ and $\mathrm{supp}(c) = \mathcal{X}$:*

$$\underbrace{\arg\max_{q \in \mathcal{Q}} \left\{ \min_{\mathbf{X}^M} \mathcal{L}_{\mathbf{X}^M, \mathbf{X}^D}(q) \right\}}_{\textit{Adversarial}} = \underbrace{\arg\max_{q \in \mathcal{Q}} \left\{ \mathbb{E}_{\mathbf{X}^M \sim c} \mathcal{L}_{\mathbf{X}^M, \mathbf{X}^D}(q) \right\}}_{\textit{Sampling-Based}} = p(f|\mathcal{D}). \tag{10}$$

The proof is given in Appendix B.2. While it is usually impractical for the measurement set to contain all the training inputs, it is still reassuring that a proper lower bound can be obtained with a finite measurement set.

## 3.3 KL DIVERGENCE GRADIENTS

While the likelihood term of the fELBO is tractable, the KL divergence term remains intractable because we don't have an explicit formula for the variational posterior density $q_\phi(\mathbf{f}^{\mathbf{X}})$. (Even if $q_\phi$ is chosen to have a tractable density in weight space (Louizos & Welling, 2017), the marginal distribution over $\mathbf{f}^{\mathbf{X}}$ is likely intractable.) To derive an approximation, we first observe that

$$\nabla_\phi \mathrm{KL}[q_\phi(\mathbf{f}^{\mathbf{X}}) \| p(\mathbf{f}^{\mathbf{X}})] = \mathbb{E}_q \left[ \nabla_\phi \log q_\phi(\mathbf{f}^{\mathbf{X}}) \right] + \mathbb{E}_\xi \left[ \nabla_\phi \mathbf{f}^{\mathbf{X}} (\nabla_{\mathbf{f}} \log q(\mathbf{f}^{\mathbf{X}}) - \nabla_{\mathbf{f}} \log p(\mathbf{f}^{\mathbf{X}})) \right]. \tag{11}$$

The first term (expected score function) in Equation (11) is zero, so we discard it.[1] The Jacobian $\nabla_\phi \mathbf{f}^{\mathbf{X}}$ can be exactly multiplied by a vector using backpropagation. Therefore, it remains to estimate the log-density derivatives $\nabla_{\mathbf{f}} \log q(\mathbf{f}^{\mathbf{X}})$ and $\nabla_{\mathbf{f}} \log p(\mathbf{f}^{\mathbf{X}})$.

The entropy derivative $\nabla_{\mathbf{f}} \log q(\mathbf{f}^{\mathbf{X}})$ is generally intractable. For priors with tractable marginal densities such as GPs (Rasmussen & Williams, 2006)[2] and Student-t Processes (Shah et al., 2014), $\nabla_{\mathbf{f}} \log p(\mathbf{f}^{\mathbf{X}})$ is tractable. However, we are also interested in implicit stochastic process priors, i.e. $\nabla_{\mathbf{f}} \log p(\mathbf{f}^{\mathbf{X}})$ is also intractable. Because the SSGE (see Section 2.3) can estimate score functions for both in-distribution and out-of-distribution samples, we use it to estimate both derivative terms in all our experiments. (We compute $\nabla_{\mathbf{f}} \log p(\mathbf{f}^{\mathbf{X}})$ exactly whenever it is tractable.)

---

[1] But note that it may be useful as a control variate (Roeder et al., 2017).
[2] Appendix D.1 introduces an additional fix to deal with the GP kernel matrix stability issue.

---

**Algorithm 1** Functional Variational Bayesian Neural Networks (fBNNs)

---

**Require:** Dataset $\mathcal{D}$, variational posterior $g(\cdot)$, prior $p$ (explicit or implicit), KL weight $\lambda$.
**Require:** Sampling distribution $c$ for random measurement points.
 1: **while** $\phi$ not converged **do**
 2:     $\mathbf{X}^M \sim c; D_S \subset \mathcal{D}$                                    ▷ sample measurement points
 3:     $\mathbf{f}_i = g([\mathbf{X}^M, \mathbf{X}^{D_S}], \xi_i; \phi), \ i = 1 \cdots k.$          ▷ sample $k$ function values
 4:     $\Delta_1 = \frac{1}{k}\frac{1}{|D_s|}\sum_i \sum_{(x,y)} \nabla_\phi \log p(y|\mathbf{f}_i(x))$     ▷ compute log likelihood gradients
 5:     $\Delta_2 = \mathrm{SSGE}(p, \mathbf{f}_{1:k})$                          ▷ estimate KL gradients
 6:     $\phi \leftarrow \mathrm{Optimizer}(\phi, \Delta_1 - \lambda\Delta_2)$             ▷ update the parameters
 7: **end while**

---

### 3.4 THE ALGORITHM

Now we present the whole algorithm for fBNNs in Algorithm 1. In each iteration, our measurement points include a mini-batch $\mathcal{D}_s$ from the training data and random points $\mathbf{X}^M$ from a distribution $c$. We forward $\mathbf{X}^{D_s}$ and $\mathbf{X}^M$ together through the network $g(\cdot; \phi)$ which defines the variational posterior $q_\phi$. Then we try to maximize the following objective corresponding to fELBO:

$$\frac{1}{|\mathcal{D}_s|}\sum_{(\mathbf{x},y)\in\mathcal{D}_s} \mathrm{E}_{q_\phi}\left[\log p(y|f(\mathbf{x}))\right] - \lambda\mathrm{KL}[q(\mathbf{f}^{\mathcal{D}_s}, \mathbf{f}^M)\|p(\mathbf{f}^{\mathcal{D}_s}, \mathbf{f}^M)]. \tag{12}$$

Here $\lambda$ is a regularization hyperparameter. In principle, $\lambda$ should be set as $\frac{1}{|\mathcal{D}|}$ to match fELBO in Equation (5). However, because the KL in Equation (12) uses a restricted number of measurement points, it only terms a lower bound of the functional KL divergence $\mathrm{KL}[q(f)\|p(f)]$, thus bigger $\lambda$ is favored to control overfitting. We used $\lambda = \frac{1}{|\mathcal{D}_s|}$ in practice, in which case Equation (12) is a proper lower bound of $\log p(\mathcal{D}_s)$, as shown in Theorem 2. Moreover, when using GP priors, we injected Gaussian noise on the function outputs for stability consideration (see Appendix D.1 for details).

## 4 RELATED WORK

**Bayesian neural networks.**    Variational inference was first applied to neural networks by Peterson (1987) and Hinton & Van Camp (1993). More recently, Graves (2011) proposed a practical method for variational inference with fully factorized Gaussian posteriors which used a simple (but biased) gradient estimator. Improving on that work, Blundell et al. (2015) proposed an unbiased gradient estimator using the reparameterization trick of Kingma & Welling (2013). There has also been much work (Louizos & Welling, 2016; Sun et al., 2017; Zhang et al., 2018; Bae et al., 2018) on modelling the correlations between weights using more complex Gaussian variational posteriors. Some non-Gaussian variational posteriors have been proposed, such as multiplicative normalizing flows (Louizos & Welling, 2017) and implicit distributions (Shi et al., 2018a). Neural networks with dropout were also interpreted as BNNs (Gal & Ghahramani, 2016; Gal et al., 2017). Local reparameterization trick (Kingma et al., 2015) and Flipout (Wen et al., 2018) try to decorrelate the gradients within a mini-batch for reducing variances during training. However, all these methods place priors over the network parameters. Often, spherical Gaussian priors are placed over the weights for convenience. Other priors, including log-uniform priors (Kingma et al., 2015; Louizos et al., 2017) and horseshoe priors (Ghosh et al., 2018; Louizos et al., 2017), were proposed for specific purposes such as model compression and model selection. But the relationships of weight-space priors to the functions computed by networks are difficult to characterize.

**Functional Priors.**    There have been other recent attempts to train BNNs in the spirit of functional priors. Flam-Shepherd et al. (2017) trained a BNN prior to mimic a GP prior, but they still required variational inference in weight space. Noise Contrastive Priors (Hafner et al., 2018) are somewhat similar in spirit to our work in that they use a random noise prior in the function space. The prior is incorporated by adding a regularization term to the weight-space ELBO, and is not rich enough to encourage extrapolation and pattern discovery. Neural Processes (NP) (Garnelo et al., 2018) try to model any conditional distribution given arbitrary data points, whose prior is specified implicitly by prior samples. However, in high dimensional spaces, conditional distributions become increasingly

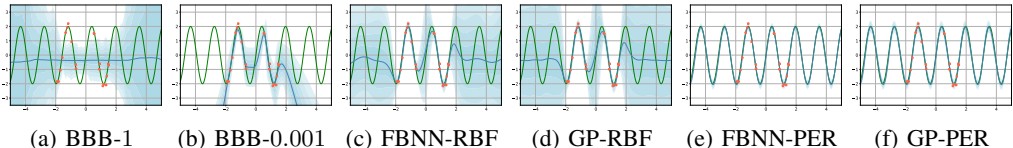

|          |          |          |          |          |          |
|:--------:|:--------:|:--------:|:--------:|:--------:|:--------:|
| (a) BBB-1 | (b) BBB-0.001 | (c) FBNN-RBF | (d) GP-RBF | (e) FBNN-PER | (f) GP-PER |

**Figure 2:** Extrapolating periodic structure. Red dots denote 20 training points. The green and blue lines represent ground truth and mean prediction, respectively. Shaded areas correspond to standard deviations. We considered GP priors with two kernels: RBF (which does not model the periodic structure), and PER + RBF (which does). In each case, the fBNN makes similar predictions to the exact GP. In contrast, the standard BBB (BBB-1) cannot even fit the training data, while BBB with scaling down KL by 0.001 (BBB-0.001) manages to fit training data, but fails to provide sensible extrapolations.

complicated to model. Variational Implicit Processes (VIP) (Ma et al., 2018) are, in a sense, the reverse of fBNNs: they specify BNN priors and use GPs to approximate the posterior. But the use of BNN priors means they can't exploit richly structured GP priors or other stochastic processes.

**Scalable Gaussian Processes.**   Gaussian processes are difficult to apply exactly to large datasets since the computational requirements scale as $O(N^3)$ time, and as $O(N^2)$ memory, where $N$ is the number of training cases. Multiple approaches have been proposed to reduce the computational complexity. However, sparse GP methods (Lázaro-Gredilla et al., 2010; Snelson & Ghahramani, 2006; Titsias, 2009; Hensman et al., 2013; 2015; Krauth et al., 2016) still suffer for very large dataset, while random feature methods (Rahimi & Recht, 2008; Le et al., 2013) and KISS-GP (Wilson & Nickisch, 2015; Izmailov et al., 2017) must be hand-tailored to a given kernel.

## 5 EXPERIMENTS

Our experiments had two main aims: (1) to test the ability of fBNNs to extrapolate using various structural motifs, including both implicit and explicit priors, and (2) to test if they perform competitively with other BNNs on standard benchmark tasks such as regression and contextual bandits.

In all of our experiments, the variational posterior is represented as a stochastic neural network with independent Gaussian distributions over the weights, i.e. $q(\mathbf{w}) = \mathcal{N}(\mathbf{w}; \boldsymbol{\mu}, \mathrm{diag}(\boldsymbol{\sigma}^2))$.[3] We always used the ReLU activation function unless otherwise specified. Measurement points were sampled uniformly from a rectangle containing the training inputs. More precisely, each coordinate was sampled from the interval $[x_{\min} - \frac{d}{2}, x_{\max} + \frac{d}{2}]$, where $x_{\min}$ and $x_{\max}$ are the minimum and maximum input values along that coordinate, and $d = x_{\max} - x_{\min}$. For experiments where we used GP priors, we first fit the GP hyperparameters to maximize the marginal likelihood on subsets of the training examples, and then fixed those hyperparameters to obtain the prior for the fBNNs.

### 5.1 EXTRAPOLATION USING STRUCTURED PRIORS

Making sensible predictions outside the range of the observed data requires exploiting the underlying structure. In this section, we consider some illustrative examples where fBNNs are able to use structured priors to make sensible extrapolations. Appendix C.2 also shows the extrapolation of fBNNs for a time-series problem.

### 5.1.1 LEARNING PERIODIC STRUCTURES

Gaussian processes can model periodic structure using a periodic kernel plus a RBF kernel:

$$k(x, x') = \sigma_1^2 \exp\left\{ -\frac{2\sin^2(\pi|x - x'|/p)}{l_1^2} \right\} + \sigma_2^2 \exp\left( -\frac{(x - x')^2}{2l_2^2} \right) \qquad (13)$$

where $p$ is the period. In this experiment, we consider 20 inputs randomly sampled from the interval $[-2, -0.5] \cup [0.5, 2]$, and targets $y$ which are noisy observations of a periodic function: $y = 2 * \sin(4x) + \epsilon$ with $\epsilon \sim \mathcal{N}(0, 0.04)$. We compared our method with Bayes By Backprop (BBB) (Blundell et al., 2015) (with a spherical Gaussian prior on $\mathbf{w}$) and Gaussian Processes. For

---

[3] One could also use stochastic activations, but we did not find that this gave any improvement.

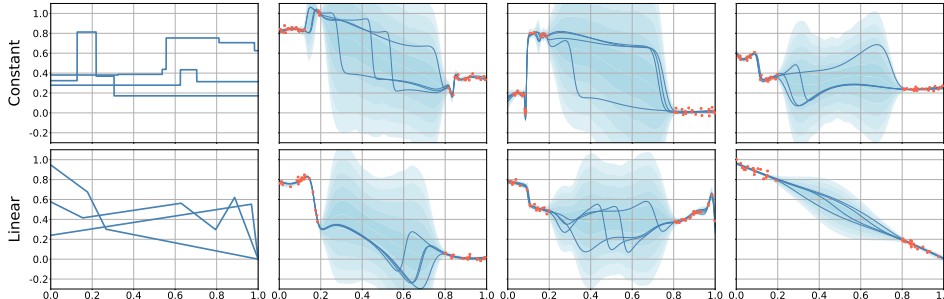

**Figure 3:** Implicit function priors and fBNN approximate posteriors. The leftmost column shows 3 prior samples. The other three columns show independent runs of the experiment. The red dots denote 40 training samples. We plot 4 posterior samples and show multiples of the predictive standard derivation as shaded areas.

fBNNs and GPs, we considered both a single RBF kernel (which does not capture the periodic structure) and PER + RBF as in eq. (13) (which does).[4]

As shown in Fig. 2, BBB failed to fit the training data, let alone recover the periodic pattern (since its prior does not encode any periodic structure). For this example, we view the GP with PER + RBF as the gold standard, since its kernel structure is designed to model periodic functions. Reassuringly, the fBNNs made very similar predictions to the GPs with the corresponding kernels, though they predicted slightly smaller uncertainty. We emphasize that the extrapolation results from the functional prior, rather than the network architecture, which does not encode periodicity, and which is not well suited to model smooth functions due to the ReLU activation function.

### 5.1.2 IMPLICIT PRIORS

Because the KL term in the fELBO is estimated using the SSGE, an implicit variational inference algorithm (as discussed in Section 2.3), the functional prior need not have a tractable marginal density. In this section, we examine approximate posterior samples and marginals for two implicit priors: a distribution over piecewise constant functions, and a distribution over piecewise linear functions. Prior samples are shown in Figure 3; see Appendix D.2 for the precise definitions. In each run of the experiment, we first sampled a random function from the prior, and then sampled 20 points from $[0, 0.2]$ and another 20 points from $[0.8, 1]$, giving a training set of 40 data points. To make the task more difficult for the fBNN, we used the tanh activation function, which is not well suited for piecewise constant or piecewise linear functions.[5]

Posterior predictive samples and marginals are shown for three different runs in Figure 3. We observe that fBNNs made predictions with roughly piecewise constant or piecewise linear structure, although their posterior samples did not seem to capture the full diversity of possible explanations of the data. Even though the tanh activation function encourages smoothness, the network learned to generate functions with sharp transitions.

### 5.2 PREDICTIVE PERFORMANCE

### 5.2.1 SMALL SCALE DATASETS

Following previous work (Hernández-Lobato & Adams, 2015), we then experimented with standard regression benchmark datasets from the UCI collection (Asuncion & Newman, 2007). In particular, we only used the datasets with less than 2000 data points so that we could fit GP hyperparameters by

---

[4]Details: we used a BNN with five hidden layers, each with 500 units. The inputs and targets were normalized to have zero mean and unit variance. For all methods, the observation noise variance was set to the true value. We used the trained GP as the prior of our fBNNs. In each iteration, measurement points included all training examples, plus 40 points randomly sampled from $[-5, 5]$. We used a training budget of 80,000 iterations, and annealed the weighting factor of the KL term linearly from 0 to 1 for the first 50,000 iterations.

[5]Details: the standard deviation of observation noise was chosen to be 0.02. In each iteration, we took all training examples, together with 40 points randomly sampled from $[0, 1]]$. We used a fully connected network with 2 hidden layers of 100 units, and tanh activations. The network was trained for 20,000 iterations.

**Table 1:** Averaged test RMSE and log-likelihood for the regression benchmarks.

| Dataset | Test RMSE | | | Test log-likelihood | | |
|---|---|---|---|---|---|---|
| | BBB | Noisy K-FAC | FBNN | BBB | Noisy K-FAC | FBNN |
| Boston | 3.171±0.149 | 2.742±0.125 | **2.378±0.104** | -2.602±0.031 | -2.446±0.029 | **-2.301±0.038** |
| Concrete | 5.678±0.087 | 5.019±0.127 | **4.935±0.180** | -3.149±0.018 | **-3.039±0.025** | -3.096±0.016 |
| Energy | 0.565±0.018 | 0.485±0.023 | **0.412±0.017** | -1.500±0.006 | -1.421±0.005 | **-0.684±0.020** |
| Wine | 0.643±0.012 | **0.637±0.011** | 0.673±0.014 | -0.977±0.017 | **-0.969±0.014** | -1.040±0.013 |
| Yacht | 1.174±0.086 | 0.979±0.077 | **0.607±0.068** | -2.408±0.007 | -2.316±0.006 | **-1.033±0.033** |

maximizing marginal likelihood exactly. Each dataset was randomly split into training and test sets, comprising 90% and 10% of the data respectively. This splitting process was repeated 10 times to reduce variability.[6]

We compared our fBNNs with Bayes By Backprop (BBB) (Blundell et al., 2015) and Noisy K-FAC (Zhang et al., 2018). In accordance with Zhang et al. (2018), we report root mean square error (RMSE) and test log-likelihood. The results are shown in Table 1. On most datasets, our fBNNs outperformed both BBB and NNG, sometimes by a significant margin.

### 5.2.2 LARGE SCALE DATASETS

Observe that fBNNs are naturally scalable to large datasets because they access the data only through the expected log-likelihood term, which can be estimated stochastically. In this section, we verify this experimentally. We compared fBNNs and BBB with large scale UCI datasets, including *Naval*, *Protein Structures*, *Video Transcoding (Memory, Time)* and *GPU kernel performance*. We randomly split the datasets into 80% training, 10% validation, and 10% test. We used the validating set to select the hyperparameters and performed early stopping.

**Table 2:** Averaged test RMSE and log-likelihood for the regression benchmarks.

| Dataset | N | Test RMSE | | Test log-likelihood | |
|---|---|---|---|---|---|
| | | BBB | FBNN | BBB | FBNN |
| Naval | 11934 | 1.6E-4±0.000 | **1.2E-4±0.000** | 6.950±0.052 | **7.130±0.024** |
| Protein | 45730 | 4.331±0.033 | **4.326±0.019** | -2.892±0.007 | **-2.892±0.004** |
| Video Memory | 68784 | 1.879±0.265 | **1.858±0.036** | **-1.999±0.054** | -2.038±0.021 |
| Video Time | 68784 | 3.632±1.974 | **3.007±0.127** | **-2.390±0.040** | -2.471±0.018 |
| GPU | 241600 | 21.886±0.673 | **19.50±0.171** | -4.505±0.031 | **-4.400±0.009** |

Both methods were trained for 80,000 iterations.[7] We used 1 hidden layer with 100 hidden units for all datasets. For the prior of fBNNs, we used a GP with Neural Kernel Network (NKN) kernels as used in Sun et al. (2018). We note that GP hyperparameters were fit using mini-batches of size 1000 with 10000 iterations. In each iteration, measurement sets consist of 500 training samples and 5 or 50 points from the sampling distribution $c$, tuned by validation performance. We ran each experiment 5 times, and report the mean and standard deviation in Table 2. More large scale regression results with bigger networks can be found at Appendix C.4 and Appendix C.5.

### 5.3 CONTEXTUAL BANDITS

One of the most important applications of uncertainty modelling is to guide exploration in settings such as bandits, Bayesian optimization (BO), and reinforcement learning. In this section, we evaluate fBNNs on a recently introduced contextual bandits benchmark (Riquelme et al., 2018). In contextual bandits problems, the agent tries to select the action with highest reward given some input context. Because the agent learns about the model gradually, it should balance between exploration

---

[6]Details: For all datasets, we used networks with one hidden layer of 50 hidden units. We first fit GP hyper-parameters using marginal likelihood with a budget of 10,000 iterations. We then trained the observation variance and kept it lower bounded by GP observation variance. FBNNs were trained for 2,000 epochs. And in each iteration, measurement points included 20 training examples, plus 5 points randomly sampled.

[7]We tune the learning rate from $[0.001, 0.01]$. We tuned between not annealing the learning rate or annealing it by 0.1 at 40000 iterations. We evaluated the validating set in each epoch, and selected the epoch for testing based on the validation performance. To control overfitting, we used Gamma$(6., 6.)$ prior following (Hernández-Lobato & Adams, 2015) for modelling observation precision and perform inference.

**Table 3:** Contextual bandits regret. Results are relative to the cumulative regret of the Uniform algorithm. Numbers after the algorithm are the network sizes. We report the mean and standard derivation over 10 trials.

| | M. RANK | M. VALUE | MUSHROOM | STATLOG | COVERTYPE | FINANCIAL | JESTER | ADULT |
|---|---|---|---|---|---|---|---|---|
| FBNN $1 \times 50$ | 4.7 | 41.9 | $21.38 \pm 7.00$ | $8.85 \pm 4.55$ | $47.16 \pm 2.39$ | $9.90 \pm 2.40$ | $75.55 \pm 5.51$ | $\mathbf{88.43 \pm 1.95}$ |
| FBNN $2 \times 50$ | 6.5 | 43.0 | $24.57 \pm 10.81$ | $10.08 \pm 5.66$ | $49.04 \pm 3.75$ | $11.83 \pm 2.95$ | $73.85 \pm 6.82$ | $88.81 \pm 3.29$ |
| FBNN $3 \times 50$ | 7 | 45.0 | $34.03 \pm 13.95$ | $7.73 \pm 4.37$ | $50.14 \pm 3.13$ | $14.14 \pm 1.99$ | $74.27 \pm 6.54$ | $89.68 \pm 1.66$ |
| FBNN $1 \times 500$ | **3.8** | 41.3 | $21.90 \pm 9.95$ | $6.50 \pm 2.97$ | $47.45 \pm 1.86$ | $\mathbf{7.83 \pm 0.77}$ | $74.81 \pm 5.57$ | $89.03 \pm 1.78$ |
| FBNN $2 \times 500$ | 4.2 | 41.2 | $23.93 \pm 11.59$ | $7.98 \pm 3.08$ | $46.00 \pm 2.01$ | $10.67 \pm 3.52$ | $\mathbf{68.88 \pm 7.09}$ | $89.70 \pm 2.01$ |
| FBNN $3 \times 500$ | 4.2 | **40.9** | $19.07 \pm 4.97$ | $10.04 \pm 5.09$ | $\mathbf{45.24 \pm 2.11}$ | $11.48 \pm 2.20$ | $69.42 \pm 7.56$ | $90.01 \pm 1.70$ |
| MULTITASKGP | 4.3 | 41.7 | $20.75 \pm 2.08$ | $7.25 \pm 1.80$ | $48.37 \pm 3.50$ | $8.07 \pm 1.13$ | $76.99 \pm 6.01$ | $88.64 \pm 3.20$ |
| BBB $1 \times 50$ | 10.8 | 52.7 | $24.41 \pm 6.70$ | $25.67 \pm 3.46$ | $58.25 \pm 5.00$ | $37.69 \pm 15.34$ | $75.39 \pm 6.32$ | $95.07 \pm 1.57$ |
| BBB $1 \times 500$ | 13.7 | 66.2 | $26.41 \pm 8.71$ | $51.29 \pm 11.27$ | $83.91 \pm 4.62$ | $57.20 \pm 7.19$ | $78.94 \pm 4.98$ | $99.21 \pm 0.79$ |
| BBALPHADIV | 15 | 83.8 | $61.00 \pm 6.47$ | $70.91 \pm 10.22$ | $97.63 \pm 3.21$ | $85.94 \pm 4.88$ | $87.80 \pm 5.08$ | $99.60 \pm 1.06$ |
| PARAMNOISE | 10 | 47.9 | $20.33 \pm 13.12$ | $13.27 \pm 2.85$ | $65.07 \pm 3.47$ | $17.63 \pm 4.27$ | $74.94 \pm 7.24$ | $95.90 \pm 2.20$ |
| NEURALLINEAR | 10.8 | 48.8 | $16.56 \pm 11.60$ | $13.96 \pm 1.51$ | $64.96 \pm 2.54$ | $18.57 \pm 2.02$ | $82.14 \pm 3.64$ | $96.87 \pm 0.92$ |
| LINFULLPOST | 8.3 | 46.0 | $14.71 \pm 0.67$ | $19.24 \pm 0.77$ | $58.69 \pm 1.17$ | $10.69 \pm 0.92$ | $77.76 \pm 5.67$ | $95.00 \pm 1.26$ |
| DROPOUT | 5.5 | 41.7 | $\mathbf{12.53 \pm 1.82}$ | $12.01 \pm 6.11$ | $48.95 \pm 2.19$ | $14.64 \pm 3.95$ | $71.38 \pm 7.11$ | $90.62 \pm 2.21$ |
| RMS | 6.5 | 43.9 | $15.29 \pm 3.06$ | $11.38 \pm 5.63$ | $58.96 \pm 4.97$ | $10.46 \pm 1.61$ | $72.09 \pm 6.98$ | $95.29 \pm 1.50$ |
| BOOTRMS | 4.7 | 42.6 | $18.05 \pm 11.20$ | $\mathbf{6.13 \pm 1.03}$ | $53.63 \pm 2.15$ | $8.69 \pm 1.30$ | $74.71 \pm 6.00$ | $94.18 \pm 1.94$ |
| UNIFORM | 16 | 100 | $100.0 \pm 0.0$ | $100.0 \pm 0.0$ | $100.0 \pm 0.0$ | $100.0 \pm 0.0$ | $100.0 \pm 0.0$ | $100.0 \pm 0.0$ |

and exploitation to maximize the cumulative reward. Thompson sampling (Thompson, 1933) is one promising approach which repeatedly samples from the posterior distribution over parameters, choosing the optimal action according to the posterior sample.

We compared our fBNNs with the algorithms benchmarked in (Riquelme et al., 2018). We ran the experiments for all algorithms and tasks using the default settings open sourced by Riquelme et al. (2018). For fBNNs, we kept the same settings, including batchsize (512), training epochs (100) and training frequency (50). For the prior, we use the multi-task GP of Riquelme et al. (2018). Measurement sets consisted of training batches, combined with 10 points sampled from data regions. We ran each experiment 10 times; the mean and standard derivation are reported in Table 3 (Appendix C.1 has the full results for all experiments.). Similarly to Riquelme et al. (2018), we also report the mean rank and mean regret.

As shown in Table 3, fBNNs outperformed other methods by a wide margin. Additionally, fBNNs maintained consistent performance even with deeper and wider networks. By comparison, BBB suffered significant performance degradation when the hidden size was increased from 50 to 500. This is consistent with our hypothesis that functional variational inference can gracefully handle networks with high capacity.

## 5.4 BAYESIAN OPTIMIZATION

Another domain where efficient exploration requires accurate uncertainty modeling is Bayesian optimization. Our experiments with Bayesian optimization are described in App C.3. We compared BBB, RBF Random Feature (Rahimi & Recht, 2008) and our fBNNs in the context of Max-value Entropy Search (MES) (Wang & Jegelka, 2017), which requires explicit function samples for Bayesian Optimization. We performed BO over functions sampled from Gaussian Processes corresponding to RBF, Matern12 and ArcCosine kernels, and found our fBNNs achieved comparable or better performance than RBF Random Feature.

## 6 CONCLUSIONS

In this paper we investigated variational inference between stochastic processes. We proved that the KL divergence between stochastic processes equals the supremum of KL divergence for marginal distributions over all finite measurement sets. Then we presented two practical functional variational inference approaches: adversarial and sampling-based. Adopting BNNs as the variational posterior yields our functional variational Bayesian neural networks. Empirically, we demonstrated that fBNNs extrapolate well over various structures, estimate reliable uncertainties, and scale to large datasets.

## ACKNOWLEDGEMENTS

We thank Ricky Chen, Kevin Luk and Xuechen Li for their helpful comments on this project. SS was supported by a Connaught New Researcher Award and a Connaught Fellowship. GZ was supported by an MRIS Early Researcher Award. RG acknowledges funding from the CIFAR Canadian AI Chairs program.

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

## A  FUNCTIONAL KL DIVERGENCE

### A.1  BACKGROUND

We begin with some basic terminology and classical results. See Gray (2011) and Folland (2013) for more details.

**Definition 1** (KL divergence). *Given a probability measure space $(\Omega, \mathcal{F}, P)$ and another probability measure $M$ on the smae space, the KL divergence of $P$ with respect to $M$ is defined as*

$$\mathrm{KL}[P\|M] = \sup_{\mathcal{Q}} \mathrm{KL}[P_{\mathcal{Q}}\|M_{\mathcal{Q}}]. \tag{14}$$

*where the supremum is taken over all finite measurable partitions $\mathcal{Q} = \{Q_i\}_{i=1}^{n}$ of $\Omega$, and $P_{\mathcal{Q}}, M_{\mathcal{Q}}$ represent the discrete measures over the partition $\mathcal{Q}$, respectively.*

**Definition 2** (Pushforward measure). *Given probability spaces $(X, \mathcal{F}_X, \mu)$ and $(Y, \mathcal{F}_Y, \nu)$, we say that measure $\nu$ is a pushforward of $\mu$ if $\nu(A) = \mu(f^{-1}(A))$ for a measurable $f : X \to Y$ and any $A \in \mathcal{F}_Y$. This relationship is denoted by $\nu = \mu \circ f^{-1}$.*

**Definition 3** (Canonical projection map). *Let $T$ be an arbitrary index set, and $\{(\Omega_t, \mathcal{F}_t)\}_{t \in T}$ be some collection of measurable spaces. For each subset $J \subset I \subset T$, define $\Omega^J = \prod_{t \in J} \Omega_t$. We call $\pi_{I \to J}$ the canonical projection map from $I$ to $J$ if*

$$\pi_{I \to J}(w) = w|_J \in \Omega^J, \forall w \in \Omega^I. \tag{15}$$

*Where $w|_J$ is defined as, if $w = (w_i)_{i \in I}$, then $w|_J = (w_i)_{i \in J}$.*

**Definition 4** (Cylindrical $\sigma$-algebra). *Let $T$ be an arbitrary index set, $(\Omega, \mathcal{F})$ be a measurable space. Suppose*

$$\Omega^T = \{f : f(t) \in \Omega, t \in T\}. \tag{16}$$

*is the set of $\Omega$-valued functions. A cylinder subset is a finitely restricted set defined as*

$$C_{t_1, \cdots, t_n}(B_1, \cdots, B_n) = \{f \in \Omega^T : f(t_i) \in B_i, 1 \leq i \leq n\}. \tag{17}$$

*Let*

$$\mathcal{G}_{t_1, \cdots, t_n} = \{C_{t_1, \cdots, t_n}(B_1, \cdots, B_n) : B_i \in \mathcal{F}, 1 \leq i \leq n\}$$
$$\mathcal{G}_{\Omega^T} = \bigcup_{n=1}^{\infty} \bigcup_{t_i \in T, i \leq n} \mathcal{G}_{t_1, \cdots, t_n} \tag{18}$$

*We call the $\sigma$-algebra $\mathcal{F}^T := \sigma(\mathcal{G}_{\Omega^T})$ as the cylindrical $\sigma$-algebra of $\Omega^T$, and $(\Omega^T, \mathcal{F}^T)$ the cylindrical measurable space.*

The Kolmogorov Extension Theorem is the foundational result used to construct many stochastic processes, such as Gaussian processes. A particularly relevant fact for our purposes is that this theorem defines a measure on a cylindrical measurable space, using only canonical projection measures on finite sets of points.

**Theorem 4** (Kolmogorov extension theorem (Øksendal, 2003)). *Let $T$ be an arbitrary index set. $(\Omega, \mathcal{F})$ is a standard measurable space, whose cylindrical measurable space on $T$ is $(\Omega^T, \mathcal{F}^T)$. Suppose that for each finite subset $I \subset T$, we have a probability measure $\mu_I$ on $\Omega^I$, and these measures satisfy the following compatibility relationship: for each subset $J \subset I$, we have*

$$\mu_J = \mu_I \circ \pi_{I \to J}^{-1}. \tag{19}$$

*Then there exists a unique probability measure $\mu$ on $\Omega^T$ such that for all finite subsets $I \subset T$,*

$$\mu_I = \mu \circ \pi_{T \to I}^{-1}. \tag{20}$$

In the context of Gaussian processes, $\mu$ is a Gaussian measure on a separable Banach space, and the $\mu_I$ are marginal Gaussian measures at finite sets of input positions (Mallasto & Feragen, 2017).

**Theorem 5.** *Suppose that $M$ and $P$ are measures on the sequence space corresponding to outcomes of a sequence of random variables $X_0, X_1, \cdots$ with alphabet $A$. Let $\mathcal{F}_n = \sigma(X_0, \cdots, X_{n-1})$, which asymptotically generates the $\sigma$-algebra $\sigma(X_0, X_1, \cdots)$. Then*

$$\mathrm{KL}[P\|M] = \lim_{n \to \infty} \mathrm{KL}[P_{\mathcal{F}_n}\|M_{\mathcal{F}_n}] \tag{21}$$

*Where $P_{\mathcal{F}_n}, M_{\mathcal{F}_n}$ denote the pushforward measures with $f : f(X_0, X_1, \cdots) = f(X_0, \cdots, X_{n-1})$, respectively.*

## A.2 FUNCTIONAL KL DIVERGENCE

Here we clarify what it means for the measurable sets to only depend on the values at a countable set of points. To begin with, we firstly introduce some definitions.

**Definition 5** (Replacing function). *For $f \in \Omega^T, t_0 \in T, v \in \Omega$, a replacing function $f^r_{t_0,v}$ is,*

$$f^r_{t_0,v}(t) = \begin{cases} f(t), & t \neq t_0 \\ v, & t = t_0 \end{cases}. \tag{22}$$

**Definition 6** (Restricted Indices). *For $H \in \mathcal{F}^T$, we define the free indices $\tau^c(H)$:*

$$\tau^c(H) = \{t \,|\, \textit{for any } f \in H, \textit{we have } f^r_{t,v} \in H \textit{ for all } v \in \Omega\} \tag{23}$$

*The complement $\tau(H) = T \backslash \tau^c(H)$ is the set of restricted indices. For example, for the set $H = \{f | f \in \mathbb{R}^T, f(0) \in (-1, 2), f(1) \in (0, 1)\}$, the restricted indices are $\tau(H) = \{0, 1\}$.*

Restricted index sets satisfy the following properties:

- for any $H \in \mathcal{F}^T, \tau(H) = \tau(H^c)$.
- for any indices set $I$ and measureable sets $\{H_i; H_i \in \mathcal{F}^T\}_{i \in I}, \tau(\underset{i \in I}{\cup} H_i) \subseteq \underset{i \in I}{\cup} \tau(H_i)$.

Having defined restricted indices, a key step in our proof is to show that, for any measureable set $H$ in a cylindrical measureable space $(\Omega^T, \mathcal{F}^T)$, its set of restricted indices $\tau(H)$ is countable.

**Lemma 6.** *Given a cylindrical measureable space $(\Omega^T, \mathcal{F}^T)$, for any $H \in \mathcal{F}^T, \tau(H)$ is countable.*

*Proof.* Define $\mathcal{H} = \{H | H \in \mathcal{F}^T, \tau(H) \text{ is countable}\}, \mathcal{H} \subseteq \mathcal{F}^T$. By the two properties of restricted indices in Definition 6, $\mathcal{H}$ is a $\sigma$-algebra on $\Omega^T$.

On the other hand, $\mathcal{F}^T = \sigma(\mathcal{G}_{\Omega^T})$. Because any set in $\mathcal{G}_{\Omega^T}$ has finite restricted indices, $\mathcal{G}_{\Omega^T} \subseteq \mathcal{H}$. Therefore $\mathcal{H}$ is a $\sigma$-algebra containing $\mathcal{G}_{\Omega^T}$. Thus $\mathcal{H} \supseteq \sigma(\mathcal{G}_{\Omega^T}) = \mathcal{F}^T$.

Overall, we conclude $\mathcal{H} = \mathcal{F}^T$. For any $H \in \mathcal{F}^T, \tau(H)$ is countable. $\square$

**Theorem 7.** *For two stochastic processes $P, M$ on a cylindrical measurable space $(\Omega^T, \mathcal{F}^T)$, the KL divergence of $P$ with respect to $M$ satisfies,*

$$\text{KL}[P\|M] = \sup_{T_d} \text{KL}[P_{T_d}\|M_{T_d}],$$

*where the supremum is over all finite indices subsets $T_d \subseteq T$, and $P_{T_d}, M_{T_d}$ represent the canonical projection maps $\pi_{T \to T_d}$ of $P, M$, respectively.*

*Proof.* Recall that stochastic processes are defined over a cylindrical $\sigma$-algebra $\mathcal{F}^T$. By Lemma 6, for every set $H \in \mathcal{F}^T$, the restricted index set $\tau(H)$ is countable. Our proof proceeds in two steps:

1. Any finite measurable partition of $\Omega^T$ corresponds to a finite measurable partition over some $\Omega^{T_c}$, where $T_c$ is a countable index set.

2. Correspondence between partitions implies correspondence between KL divergences.

3. KL divergences over a countable indices set can be represented as supremum of KL divergences over finite indices sets.

**Step 1.** By Definition 1,
$$\text{KL}[P\|M] = \sup_{\mathcal{Q}_{\Omega^T}} \text{KL}[P_{\mathcal{Q}_{\Omega^T}}\|M_{\mathcal{Q}_{\Omega^T}}],$$

where the sup is over all finite measurable partitions of the function space $\Omega^T$, denoted by $\mathcal{Q}_{\Omega^T}$:

$$\mathcal{Q}_{\Omega^T} = \{Q^{(1)}_{\Omega^T}, \ldots, Q^{(k)}_{\Omega^T} | \bigcup_{i=1}^k Q^{(i)}_{\Omega^T} = \Omega^T, Q^{(i)}_{\Omega^T} \in \mathcal{F}^T \text{ are disjoint sets}, k \in \mathbb{N}^+\}.$$

By Lemma 6, each $\tau(Q_{\Omega^T}^{(i)})$ is countable. So the combined restricted index set $T_c := \bigcup_{i=1}^{k} \tau(Q_{\Omega^T}^{(i)})$ is countable.

Consider the canonical projection mapping $\pi_{T \to T_c}$, which induces a partition on $\Omega^{T_c}$, denoted by $\mathcal{Q}_{\Omega^{T_c}}$:

$$Q_{\Omega^{T_c}}^{(i)} = \pi_{T \to T_c}(Q_{\Omega^T}^{(i)}).$$

The pushforward measure defined by this mapping is

$$P_{T_c} = P \circ \pi_{T \to T_c}^{-1}, \quad M_{T_c} = M \circ \pi_{T \to T_c}^{-1}.$$

**Step 2.** Then we have

$$\mathrm{KL}[P\|M] = \sup_{\mathcal{Q}_{\Omega^T}} \mathrm{KL}[P_{\mathcal{Q}_{\Omega^T}}\|M_{\mathcal{Q}_{\Omega^T}}] \tag{24}$$

$$= \sup_{\mathcal{Q}_{\Omega^T}} \sum_i P(Q_{\Omega^T}^{(i)}) \log \frac{P(Q_{\Omega^T}^{(i)})}{M(Q_{\Omega^T}^{(i)})} \tag{25}$$

$$= \sup_{T_c} \sup_{\mathcal{Q}_{\Omega^{T_c}}} \sum_i P_{T_c}(Q_{\Omega^{T_c}}^{(i)}) \log \frac{P_{T_c}(Q_{\Omega^{T_c}}^{(i)})}{M_{T_c}(Q_{\Omega^{T_c}}^{(i)})} \tag{26}$$

$$= \sup_{T_c} \mathrm{KL}[P_{T_c}\|M_{T_c}], \tag{27}$$

**Step 3.** Denote $\mathcal{D}(T_c)$ as the collection of all finite subsets of $T_c$. For any finite set $T_d \in \mathcal{D}(T_c)$, we denote $P_{T_d}$ as the pushforward measure of $P_{T_c}$ on $\Omega^{T_d}$. From the Kolmogorov Extension Theorem (Theorem 4), we know that $P_{T_d}$ corresponds to the finite marginals of $P$ at $\Omega^{T_d}$. Because $T_c$ is countable, based on Theorem 5, we have,

$$\mathrm{KL}[P\|M] = \sup_{T_c} \mathrm{KL}[P_{T_c}\|M_{T_c}]$$

$$= \sup_{T_c} \sup_{T_d \in \mathcal{D}(T_c)} \mathrm{KL}[P_{T_d}\|M_{T_d}]. \tag{28}$$

We are left with the last question: whether each $T_d$ is contained in some $\mathcal{D}(T_c)$ ?

For any finite indices set $T_d$, we build a finite measureable partition $Q$. Let $\Omega = \Omega_0 \cup \Omega_1, \Omega_0 \cap \Omega_1 = \emptyset$. Assume $|T_d| = K, T_d = \{T_d(k)\}_{k=1:K}$, let $I = \{I^i | I^i = (I_1^i, I_2^i, \cdots I_K^i)\}_{i=1:2^K}$ to be all $K$-length binary vectors. We define the partition,

$$\mathcal{Q} = \{Q^i\}_{i=1:2^K}, \tag{29}$$

$$Q^i = \bigcap_{k=1}^{K} \{f | \begin{cases} f(T_f(k)) \in \Omega_0, & I^i(k) = 0 \\ f(T_f(k)) \in \Omega_1, & I^i(k) = 1 \end{cases} \}. \tag{30}$$

Through this settting, $\mathcal{Q}$ is a finite parition of $\Omega^T$, and $T_c(\mathcal{Q}) = T_d$. Therefore $T_d$ in Equation (28) can range over all finite index sets, and we have proven the theorem.

$$\mathrm{KL}[P\|M] = \sup_{T_d} \mathrm{KL}[P_{T_d}\|M_{T_d}]. \tag{31}$$

$\square$

## A.3 KL Divergence between Conditional Stochastic Processes

In this section, we give an example of computing the KL divergence between two conditional stochastic processes. Consider two datasets $\mathcal{D}_1, \mathcal{D}_2$, the KL divergence between two conditional

stochastic processes is

$$
\begin{aligned}
\mathrm{KL}[p(f|\mathcal{D}_1)\|p(f|\mathcal{D}_2)] &= \sup_{n,\mathbf{x}_{1:n}} \mathrm{KL}[p(\mathbf{f^x}|\mathcal{D}_1)\|p(\mathbf{f^x}|\mathcal{D}_2)] \\
&= \sup_{n,\mathbf{x}_{1:n}} \mathbb{E}_{p(\mathbf{f^x},\mathbf{f}^{D_1\cup D_2}|\mathcal{D}_1)} \log \frac{p(\mathbf{f^x},\mathbf{f}^{D_1\cup D_2}|\mathcal{D}_1)}{p(\mathbf{f^x},\mathbf{f}^{D_1\cup D_2}|\mathcal{D}_2)} \\
&= \sup_{n,\mathbf{x}_{1:n}} \mathbb{E}_{p(\mathbf{f^x},\mathbf{f}^{D_1\cup D_2}|\mathcal{D}_1)} \log \frac{p(\mathbf{f}^{D_1\cup D_2}|\mathcal{D}_1)p(\mathbf{f^x}|\mathbf{f}^{D_1\cup D_2},\mathcal{D}_1)}{p(\mathbf{f}^{D_1\cup D_2}|\mathcal{D}_2)p(\mathbf{f^x}|\mathbf{f}^{D_1\cup D_2},\mathcal{D}_2)} \\
&= \sup_{n,\mathbf{x}_{1:n}} \mathbb{E}_{p(\mathbf{f^x},\mathbf{f}^{D_1\cup D_2}|\mathcal{D}_1)} \log \frac{p(\mathbf{f}^{D_1\cup D_2}|\mathcal{D}_1)p(\mathbf{f^x}|\mathbf{f}^{D_1\cup D_2})}{p(\mathbf{f}^{D_1\cup D_2}|\mathcal{D}_2)p(\mathbf{f^x}|\mathbf{f}^{D_1\cup D_2})} \\
&= \mathrm{KL}[p(\mathbf{f}^{D_1\cup D_2}|\mathcal{D}_1)\|p(\mathbf{f}^{D_1\cup D_2}|\mathcal{D}_2)] \quad (32)
\end{aligned}
$$

Therefore, the KL divergence between these two stochastic processes equals to the marginal KL divergence on the observed locations. When $\mathcal{D}_2 = \emptyset$, $p(f|\mathcal{D}_2) = p(f)$, this shows the KL divergence between posterior process and prior process are the marginal KL divergence on observed locations.

This also justifies our usage of $M$ measurement points in the adversarial functional VI and sampling-based functional VI of Section 3.

# B ADDITIONAL PROOFS

## B.1 PROOF FOR EVIDENCE LOWER BOUND

This section provides proof for Theorem 2.

*Proof of Theorem 2.* Let $\mathbf{X}^M = \mathbf{X}\backslash\mathbf{X}^D$ be measurement points which aren't in the training data.

$$
\begin{aligned}
\mathcal{L}_{\mathbf{X}}(q) &= \mathbb{E}_q[\log p(\mathbf{y}^D|\mathbf{f}^D) + \log p(\mathbf{f^X}) - \log q_\phi(\mathbf{f^X})] \\
&= \mathbb{E}_q[\log p(\mathbf{y}^D|\mathbf{f}^D) + \log p(\mathbf{f}^D,\mathbf{f}^M) - \log q_\phi(\mathbf{f}^D,\mathbf{f}^M)] \\
&= \log p(\mathcal{D}) - \mathbb{E}_q\left[\log \frac{q_\phi(\mathbf{f}^D,\mathbf{f}^M)p(\mathcal{D})}{p(\mathbf{y}^D|\mathbf{f}^D)p(\mathbf{f}^D,\mathbf{f}^M)}\right] \\
&= \log p(\mathcal{D}) - \mathbb{E}_q\left[\log \frac{q_\phi(\mathbf{f}^D,\mathbf{f}^M)}{p(\mathbf{f}^D,\mathbf{f}^M|\mathcal{D})}\right] \\
&= \log p(\mathcal{D}) - \mathrm{KL}[q_\phi(\mathbf{f}^D,\mathbf{f}^M)\|p(\mathbf{f}^D,\mathbf{f}^M|\mathcal{D})] \quad (33)
\end{aligned}
$$

$\square$

## B.2 CONSISTENCY FOR GAUSSIAN PROCESSES

This section provides proof for consistency in Corollary 3.

*Proof of Corollary 3.* By the assumption that both $q(\mathcal{D})$ and $p(f|\mathcal{D})$ are Gaussian processes:

$$
\begin{aligned}
p(f(\cdot)|\mathcal{D}) &: \quad \mathcal{GP}(m_p(\cdot), k_p(\cdot,\cdot)), &(34) \\
q(f(\cdot)) &: \quad \mathcal{GP}(m_q(\cdot), k_q(\cdot,\cdot)), &(35)
\end{aligned}
$$

where $m$ and $k$ denote the mean and covariance functions, respectively.

In this theorem, we also assume the measurement points cover all training locations as in Equation (9), where we have (based on Theorem 2):

$$
\mathcal{L}_{\mathbf{X}}(q) = \log p(\mathcal{D}) - \mathrm{KL}[q(\mathbf{f}^D,\mathbf{f}^M)\|p(\mathbf{f}^D,\mathbf{f}^M|\mathcal{D})] \leq \log p(\mathcal{D}). \quad (36)
$$

Therefore, when the variational posterior process is sufficiently expressive and reaches its optimum, we must have $\mathrm{KL}[q(\mathbf{f}^D,\mathbf{f}^M)\|p(\mathbf{f}^D,\mathbf{f}^M|\mathcal{D})] = 0$ and thus $\mathrm{KL}[q(\mathbf{f}^M)\|p(\mathbf{f}^M|\mathcal{D})] = 0$ at $\mathbf{X}^M = [\mathbf{x}_1,\ldots,\mathbf{x}_M]^\top \in \mathcal{X}^M$, which implies

$$
\mathcal{N}(m_p(\mathbf{X}^M), k_p(\mathbf{X}^M,\mathbf{X}^M)) = \mathcal{N}(m_q(\mathbf{X}^M), k_q(\mathbf{X}^M,\mathbf{X}^M)). \quad (37)
$$

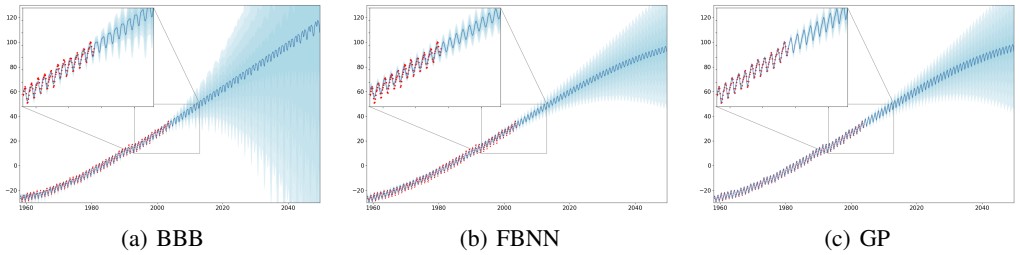

|  (a) BBB | (b) FBNN | (c) GP |

**Figure 4:** Predictions on Mauna datasets. Red dots are training points. The blue line is the mean prediction and shaded areas correspond to standard deviations.

Here $m(\mathbf{X}) = [m(\mathbf{x}_1), \ldots, m(\mathbf{x}_M)]^\top$, and $\left[k(\mathbf{X}^M, \mathbf{X}^M)\right]_{ij} = k(\mathbf{x}_i, \mathbf{x}_j)$.

Remember that in sampling-based functional variational inference, $\mathbf{X}^M$ are randomly sampled from $c(\mathbf{x})$, and $\mathrm{supp}(c) = \mathcal{X}$. Thus when it reaches optimum, we have $\mathbb{E}_{\mathbf{X}^M \sim c}\mathrm{KL}[q(\mathbf{f}^M)\|p(\mathbf{f}^M|\mathcal{D})] = 0$. Because the KL divergence is always non-negative, we have that $\mathrm{KL}[q(\mathbf{f}^M)\|p(\mathbf{f}^M|\mathcal{D})] = 0$ for any $\mathbf{X}^M \in \mathcal{X}^M$. For adversarial functional variational inference, this is also obvious due to $\sup_{\mathbf{X}^M} \mathrm{KL}[q(\mathbf{f}^M)\|p(\mathbf{f}^M|\mathcal{D})] = 0$.

So we have that Equation (37) holds for any $\mathbf{X}^M \in \mathcal{X}^M$. Given $M > 1$, then for $\forall 1 \le i < j \le M$, we have $m_p(\mathbf{x}_i) = m_q(\mathbf{x}_i)$, and $k_p(\mathbf{x}_i, \mathbf{x}_j) = k_q(\mathbf{x}_i, \mathbf{x}_j)$, which implies

$$m_p(\cdot) = m_q(\cdot), \quad k_p(\cdot, \cdot) = k_q(\cdot, \cdot). \tag{38}$$

Because GPs are uniquely determined by their mean and covariance functions, we arrive at the conclusion. ☐

## C  ADDITIONAL EXPERIMENTS

### C.1  CONTEXTUAL BANDITS

Here we present the full table for the contextual bandits experiment.

**Table 4:** Contextual bandits regret. Results are relative to the cumulative regret of the Uniform algorithm. Numbers after the algorithm are the network sizes. We report the mean and standard derivation over 10 trials.

|  | M. RANK | M. VALUE | MUSHROOM | STATLOG | COVERTYPE | FINANCIAL | JESTER | ADULT | CENSUS | WHEEL |
|---|---|---|---|---|---|---|---|---|---|---|
| FBNN 1 × 50 | 5.875 | 46.0 | 21.38 ± 7.00 | 8.85 ± 4.55 | 47.16 ± 2.39 | 9.90 ± 2.40 | 75.55 ± 5.51 | **88.43 ± 1.95** | 51.43 ± 2.34 | 65.05 ± 20.10 |
| FBNN 2 × 50 | 7.125 | 47.0 | 24.57 ± 10.81 | 10.08 ± 5.66 | 49.04 ± 3.75 | 11.83 ± 2.95 | 73.85 ± 6.82 | 88.81 ± 3.29 | 50.09 ± 2.74 | 67.76 ± 25.74 |
| FBNN 3 × 50 | 8.125 | 48.9 | 34.03 ± 13.95 | 7.73 ± 4.37 | 50.14 ± 3.13 | 14.14 ± 1.99 | 74.27 ± 6.54 | 89.68 ± 1.66 | 52.37 ± 3.03 | 68.60 ± 22.24 |
| FBNN 1 × 500 | **4.875** | **45.3** | 21.90 ± 9.95 | 6.50 ± 2.97 | 47.45 ± 1.86 | **7.83 ± 0.77** | 74.81 ± 5.57 | 89.03 ± 1.78 | 50.73 ± 1.53 | 63.77 ± 25.80 |
| FBNN 2 × 500 | **5.0** | **44.2** | 23.93 ± 11.59 | 7.98 ± 3.08 | 46.00 ± 2.01 | 10.67 ± 3.52 | **68.88 ± 7.09** | 89.70 ± 2.01 | 51.87 ± 2.38 | 54.57 ± 32.92 |
| FBNN 3 × 500 | 4.75 | 44.6 | 19.07 ± 4.97 | 10.04 ± 5.09 | **45.24 ± 2.11** | 11.48 ± 2.20 | 69.42 ± 7.56 | 90.01 ± 1.70 | **49.73 ± 1.35** | 61.57 ± 21.73 |
| MULTITASKGP | 5.875 | 46.5 | 20.75 ± 2.08 | 7.25 ± 1.80 | 48.37 ± 3.50 | 8.07 ± 1.13 | 76.99 ± 6.01 | 88.64 ± 3.20 | 57.86 ± 8.19 | 64.15 ± 27.08 |
| BBB 1 × 50 | 11.5 | 56.6 | 24.41 ± 6.70 | 25.67 ± 3.46 | 58.25 ± 5.00 | 37.69 ± 15.34 | 75.39 ± 6.32 | 95.07 ± 1.57 | 63.96 ± 3.95 | 72.37 ± 16.87 |
| BBB 1 × 500 | 13.375 | 68.1 | 26.41 ± 8.71 | 51.29 ± 11.27 | 83.91 ± 4.62 | 57.20 ± 7.19 | 78.94 ± 4.98 | 99.21 ± 0.79 | 92.73 ± 9.13 | 55.09 ± 13.82 |
| BBALPHADIV | 16.0 | 87.4 | 61.00 ± 6.47 | 70.91 ± 10.22 | 97.63 ± 3.21 | 85.94 ± 4.88 | 87.80 ± 5.08 | 99.60 ± 1.06 | 100.41 ± 1.54 | 95.75 ± 12.31 |
| PARAMNOISE | 10.125 | 53.0 | 20.33 ± 13.12 | 13.27 ± 2.85 | 65.07 ± 3.47 | 17.63 ± 4.27 | 74.94 ± 7.24 | 95.90 ± 2.20 | 82.67 ± 3.86 | 54.38 ± 16.20 |
| NEURALLINEAR | 10.375 | 52.3 | 16.56 ± 11.60 | 13.96 ± 1.51 | 64.96 ± 2.54 | 18.57 ± 2.02 | 82.14 ± 3.64 | 96.87 ± 0.92 | 78.94 ± 1.87 | 46.26 ± 8.40 |
| LINFULLPOST | 9.25 | — | 14.71 ± 0.67 | 19.24 ± 0.77 | 58.69 ± 1.17 | 10.69 ± 0.92 | 77.76 ± 5.67 | 95.00 ± 1.26 | CRASH | **33.88 ± 15.15** |
| DROPOUT | 7.625 | 48.3 | **12.53 ± 1.82** | 12.01 ± 6.11 | 48.95 ± 2.19 | 14.64 ± 3.95 | 71.38 ± 7.11 | 90.62 ± 2.21 | 58.53 ± 2.35 | 77.46 ± 27.58 |
| RMS | 8.875 | 53.0 | 15.29 ± 3.06 | 11.38 ± 5.63 | 58.96 ± 4.97 | 10.46 ± 1.61 | 72.09 ± 6.98 | 95.29 ± 1.50 | 85.29 ± 5.85 | 75.62 ± 30.43 |
| BOOTRMS | 7.5 | 51.9 | 18.05 ± 11.20 | **6.13 ± 1.03** | 53.63 ± 2.15 | 8.69 ± 1.30 | 74.71 ± 6.00 | 94.18 ± 1.94 | 82.27 ± 1.84 | 77.80 ± 29.55 |
| UNIFORM | 16.75 | 100 | 100.0 ± 0.0 | 100.0 ± 0.0 | 100.0 ± 0.0 | 100.0 ± 0.0 | 100.0 ± 0.0 | 100.0 ± 0.0 | 100.0 ± 0.0 | 100.0 ± 0.0 |

### C.2  TIME-SERIES EXTRAPOLATION

Besides the toy experiments, we would like to examine the extrapolation behavior of our method on real-world datasets. Here we consider a classic time-series prediction problem concerning the concentration of $CO_2$ in the atmosphere at the Mauna Loa Observatory, Hawaii (Rasmussen & Williams, 2006). The training data is given from 1958 to 2003 (with some missing values). Our goal is to model the prediction for an equally long period after 2003 (2004-2048). In Figure 4 we draw the prediction results given by BBB, fBNN, and GP. We used the same BNN architecture for BBB and fBNN: a ReLU network with 2 hidden layers, each with 100 units, and the input is a normalized year number augmented by its sin transformation, whose period is set to be one year. This special design

allows both BBB and fBNN to fit the periodic structure more easily. Both models are trained for 30k iterations by the Adam optimizer, with learning rate 0.01 and batch size 20. For fBNN the prior is the same as the GP experiment, whose kernel is a combination of RBF, RBF×PER (period set to one year), and RQ kernels, as suggested in Rasmussen & Williams (2006). Measurement points include 20 training samples and 10 points sampled from $\mathbb{U}[1958, 2048]$, and we jointly train the prior GP hyperparameters with fBNN.

In Figure 4 we could see that the performance of fBNN closely matches the exact prediction by GP. Both of them give visually good extrapolation results that successfully model the long-term trend, local variations, and periodic structures. In contrast, weight-space prior and inference (BBB) neither captures the right periodic structure, nor does it give meaningful uncertainty estimates.

## C.3 BAYESIAN OPTIMIZATION

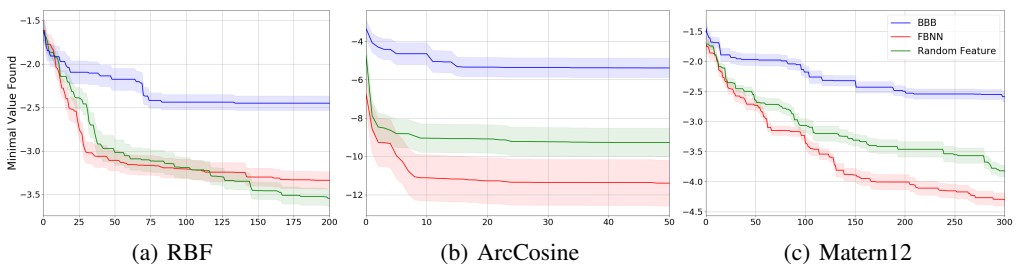

(a) RBF          (b) ArcCosine          (c) Matern12

**Figure 5:** Bayesian Optimization. We plot the minimal value found along iterations. We compare fBNN, BBB and Random Feature methods for three kinds of functions corresponding to RBF, Order-1 ArcCosine and Matern12 GP kernels. We plot mean and 0.2 standard derivation over 10 independent runs.

In this section, we adopt Bayesian Optimization to explore the advantage of coherent posteriors. Specifically, we use Max Value Entropy Search (MES) (Wang & Jegelka, 2017), which tries to maximize the information gain about the minimum value $y^\star$,

$$\alpha_t(\mathbf{x}) = \mathbb{H}(p(y|D_t, \mathbf{x})) - \mathbb{H}(p(y|D_t, \mathbf{x}, y^\star)) \approx \frac{1}{K}\sum_{y^\star}[\frac{\gamma_{y^\star}(\mathbf{x})\phi(\gamma_{y^\star}(\mathbf{x}))}{\Psi(\gamma_{y^\star}(\mathbf{x}))} - \log(\Psi(\gamma_{y^\star}(\mathbf{x})))]$$

Where $\phi$ and $\Psi$ are probability density function and cumulative density function of a standard normal distribution, respectively. The $y^\star$ is the minimum of a random function from the posterior, and $\gamma_{y^\star}(\mathbf{x}) = \frac{\mu_t(\mathbf{x}) - y^\star}{\sigma_t(\mathbf{x})}$.

With a probabilistic model, we can compute or estimate the mean $\mu_t(\mathbf{x})$ and the standard deviation $\sigma_t(\mathbf{x})$. However, to compute the MES acquisition function, samples $y^\star$ of function minima are required as well, which leads to difficulties. Typically when we model the data with a GP, we can get the posterior on a specific set of points but we don't have access to the extremes of the underlying function. In comparison, if the function posterior is represented in a parametric form, we can perform gradient decent easily and search for the minima.

We use 3-dim functions sampled from some Gaussian process prior for Bayesian optimization. Concretely, we experiment with samples from RBF, Order-1 ArcCosine and Matern12 kernels. We compare three parametric approaches: fBNN, BBB and Random Feature (Rahimi & Recht, 2008). For fBNN, we use the true kernel as functional priors. In contrast, ArcCosine and Matern12 kernels do not have simple explicit random feature expressions, therefore we use RBF random features for all three kernels. When looking for minima, we sample 10 $y^\star$. For each $y^\star$, we perform gradient descent along the sampled parametric function posterior with 30 different starting points. We use 500 dimensions for random feature. We use network with $5 \times 100$ for fBNN. For BBB, we select the network within $1 \times 100, 3 \times 100$. Because of the similar issue in Figure 1, using larger networks won't help for BBB. We use batch size 30 for both fBNN and BBB. The measurement points contain 30 training points and 30 points uniformly sampled from the known input domain of functions. For training, we rescale the inputs to $[0, 1]$, and we normalize outputs to have zero mean and unit variance. We train fBNN and BBB for 20000 iterations and anneal the coefficient of log likelihood term linearly from 0 to 1 for the first 10000 iterations. The results with 10 runs are shown in Figure 5.

As seen from Figure 5, fBNN and Random feature outperform BBB by a large margin on all three functions. We also observe fBNN performs slightly worse than random feature in terms of RBF priors. Because random feature method is exactly a GP with RBF kernel asymptotically, it sets a high standard for the parametric approaches. In contrast, fBNN outperforms random feature for both ArcCosine and Matern12 functions. This is because of the big discrepancy between such kernels and RBF random features. Because fBNN use true kernels, it models the function structures better. This experiment highlights a key advantage of fBNN, that fBNN can learn parametric function posteriors for various priors.

## C.4 VARYING DEPTH

**Table 5:** Averaged test RMSE and log-likelihood for the regression benchmarks. We compared BBB, fBNNs and VFE. The numbers $a \times b$ represent networks with $a$ hidden layers of $b$ units.

| Dataset | N | Test RMSE | | | Test log-likelihood | | |
|---|---|---|---|---|---|---|---|
| | | BBB | FBNN | VFE | BBB | FBNN | VFE |
| Kin8nm ($1\times100$) | 8192 | 0.082±0.001 | 0.079±0.001 | 0.071±0.001 | 1.082±0.008 | 1.112±0.007 | 1.241±0.005 |
| Kin8nm ($2\times100$) | 8192 | 0.074±0.001 | 0.075±0.001 | 0.071±0.001 | 1.191±0.006 | 1.151±0.007 | 1.241±0.005 |
| Kin8nm ($5\times500$) | 8192 | 0.266±0.003 | 0.076±0.001 | 0.071±0.001 | -0.279±0.007 | 1.144±0.008 | 1.241±0.005 |
| Power Plant ($1\times100$) | 9568 | 4.127±0.057 | 4.099±0.051 | 3.092±0.052 | -2.837±0.013 | -2.833±0.012 | -2.531±0.018 |
| Power Plant ($2\times100$) | 9568 | 4.081±0.054 | 3.830±0.055 | 3.092±0.052 | -2.826±0.013 | -2.763±0.013 | -2.531±0.018 |
| Power Plant ($5\times500$) | 9568 | 17.166±0.099 | 3.542±0.054 | 3.092±0.052 | -4.286±0.007 | -2.691±0.016 | -2.531±0.018 |

To compare with Variational Free Energy (VFE) (Titsias, 2009), we experimented with two medium-size datasets so that we can afford to use VFE with full batch. For VFE, we used 1000 inducing points initialized by k-means of training point. For BBB and FBNNs, we used batch size 500 with a budget of 2000 epochs. As shown in Table 5, FBNNs performed slightly worse than VFE, but the gap became smaller as we used larger networks. By contrast, BBB totally failed with large networks (5 hidden layers with 500 hidden units each layer). Finally, we note that the gap between FBNNs and VFE diminishes if we use fewer inducing points (e.g., 300 inducing points).

## C.5 LARGE SCALE REGRESSION WITH DEEPER NETWORKS

**Table 6:** Large scale regression. BBB and FBNN used networks with 5 hidden layers of 100 units.

| Dataset | N | Test RMSE | | | Test log-likelihood | | |
|---|---|---|---|---|---|---|---|
| | | BBB | FBNN | SVGP | BBB | FBNN | SVGP |
| Naval | 11934 | 1.3E-4±0.000 | 0.7E-4±0.000 | 0.3E-4±0.000 | 6.968±0.014 | 7.237±0.009 | 8.523±0.06 |
| Protein | 45730 | 3.684±0.041 | 3.659±0.026 | 3.740±0.015 | -2.715±0.012 | -2.721±0.010 | -2.736±0.003 |
| Video Memory | 68784 | 0.984±0.074 | 0.967±0.040 | 1.417±0.234 | -1.231±0.078 | -1.337±0.058 | -1.723±0.179 |
| Video Time | 68784 | 1.056±.0.178 | 1.083±0.288 | 3.216±1.154 | -1.180±0.070 | -1.468±0.279 | -2.475±0.409 |
| GPU | 241600 | 5.136±0.087 | 4.806±0.116 | 21.287±0.571 | -2.992±0.013 | -2.973±0.019 | -4.557±0.021 |

In this section we experimented on large scale regression datasets with deeper networks. For BBB and fBNNs, we used a network with 5 hidden layers of 100 units, and kept all other settings the same as Section 5.2.2. We also compared with the stochastic variational Gaussian processes (SVGP) (Hensman et al., 2013), which provides a principled mini-batch training for sparse GP methods, thus enabling GP to scale up to large scale datasets. For SVGP, we used 1000 inducing points initialized by k-means of training points (Note we cannot afford larger size of inducing points because of the cubic computational cost). We used batch size 2000 and iterations 60000 to match the training time with fBNNs. Likewise for BNNs, we used validation set to tune the learning rate from $\{0.01, 0.001\}$. We also tuned between not annealing the learning rate or annealing it by 0.1 at 30000 iterations. We evaluated the validating set in each epoch, and selected the epoch for testing based on the validation performance. The averaged results over 5 runs are shown in Table 6.

As shown in Table 6, SVGP performs better than BBB and fBNNs in terms of the smallest *naval* dataset. However, with dataset size increasing, SVGP performs worse than BBB and fBNNs by a large margin. This stems from the limited capacity of 1000 inducing points, which fails to act as sufficient statistics for large datasets. In contrast, BNNs including BBB and fBNNs can use larger networks freely without the intractable computational cost.

# D    IMPLEMENTATION DETAILS

## D.1    INJECTED NOISES FOR GAUSSIAN PROCESS PRIORS

For Gaussian process priors, $p(\mathbf{f^X})$ is a multivariate Gaussian distribution, which has an explicit density. Therefore, we can compute the gradients $\nabla_{\mathbf{f}} \log p_\phi(\mathbf{f^X})$ analytically.

In practice, we found that the GP kernel matrix suffers from stability issues. To stabilize the gradient computation, we propose to inject a small amount of Gaussian noise on the function values, i.e., to instead estimate the gradients of $\nabla_\phi \mathrm{KL}[q_\phi * p_\gamma \| p * p_\gamma]$, where $p_\gamma = \mathcal{N}(0, \gamma^2)$ is the noise distribution. This is like the instance-noise trick that is commonly used for stabilizing GAN training (Sønderby et al., 2016). Note that injecting the noise on the GP prior is equivalent to have a kernel matrix $\mathbf{K} + \gamma^2 \mathbf{I}$. Beyond that, injecting the noise on the parametric variational posterior does not affect the reparameterization trick either. Therefore all the previous estimation formulas still apply.

## D.2    IMPLICIT PRIORS

Our method is applicable to implicit priors. We experiment with piecewise constant prior and piecewise linear prior. Concretely, we randomly generate a function $f : [0, 1] \rightarrow R$ with the specific structure. To sample piecewise functions, we first sample $n \sim \mathrm{Poisson}(3.)$, then we have $n+1$ pieces within $[0, 1]$. We uniformly sample $n$ locations from $[0, 1]$ as the changing points. For piecewise constant functions, we uniformly sample $n + 1$ values from $[0, 1]$ as the function values in each piece; For piecewise linear functions, we uniformly sample $n + 1$ values for the values at first $n + 1$ locations, we force $f(1) = 0.$. Then we connect together each piece by a straight line.

