# OpenReview forum: "FUNCTIONAL VARIATIONAL BAYESIAN NEURAL NETWORKS"
_ICLR.cc/2019/Conference_

### Official Review · AnonReviewer2 · 2018-10-28
**Interesting and timely contributions hampered by rushed submission**

**Rating:** 6
**Confidence:** 4

**Review:**

-- Paper Summary --

The primary contribution of this paper is the presentation of a novel ELBO objective for training BNNs which allows for more meaningful priors to be encoded in the model rather than the less informative weight priors featured in the literature. This is achieved by way of introducing a KL measure over stochastic processes which allows for priors to take the form of GP priors and other custom variations. Two approaches are given for training the model, one inspired by GANs, and a more practical sampling-based scheme. The performance of this training scheme is validated on a variety of synthetic and real examples, choosing Bayes by Backprop as the primary competitor. An experiment on contextual bandit exploration, and an illustrative Bayesian optimisation example  provided in the supplementary material showcase the effectiveness of this method in applications where well-calibrated uncertainty is particularly pertinent.

-- Critique --

This paper makes important strides towards giving more meaningful interpretations to priors in BNNs. To the best of my knowledge, the KL divergence between stochastic processes that gives rise to an alternate ELBO has not been featured elsewhere, making this a rather interesting contribution that is supplemented by suitable theorems both in the main text and supplementary material. The introductory commentary regarding issues faced with increasing the model capacity of BNNs is particularly interesting, and the associated motivating example showing how degeneracy is countered by fBNN is clear and effective.

The GAN-inspired optimisation scheme is also well-motivated. Although the authors understandably do not pursue that scheme due to the longer computation time incurred (rendering its use impractical), it would have been interesting to see whether the optimum found using this technique is superior to the sampling based scheme used throughout the remainder of the paper. The experimental evaluation is also very solid, striking an adequate balance between synthetic and real-world examples, while also showcasing fBNNs’ effectiveness in scenarios relying on good uncertainty quantification.

In spite of the paper’s indisputable selling points, I have several issues with some aspects of this submission. For clarity, I shall distinguish my concerns between points that I believe to be particularly important, and others which are less significant:

- Monte Carlo dropout (Gal & Ghahramani, 2016), and its extensions (such as concrete dropout), are widely-regarded as being one of the most effective approaches for interpreting BNNs. Consequently, I would have expected this method to feature as a competitor in your evaluation, yet this method does not even get a cursory mention in the text.

 - The commentary on GPs in the related work paints a dour picture of their scalability by mostly listing older papers. However, flexible models such as AutoGP (Krauth et al, 2017) have been shown to obtain very good results on large datasets without imposing restrictions on the choice of kernels.

 - The regression experiments all deal with a one-layer architecture, for which the proposed method is shown to consistently obtain better results. In order to properly assess the effectiveness of the method, I would also be interested in seeing how it compares against BBB for deeper architectures on this problem. Although the authors cite the results in Figure 1 as an indicator that BBB with more layers isn’t particularly effective, it would be nice to also see this illustrated in the cross-dataset comparison presented in Section 5.2.

 - Furthermore, given that all methods are run for a fixed number of iterations, it might be sensible  to additionally report training time along with the results in the table. This should reflect the pre-processing time required to optimise GP hyperparameters when a GP prior is used. Carrying out Cholesky decompositions for 1000x1000 matrices 10k times (as described in Section 5.2.2) does not sound insignificant.

- The observation regarding the potential instability of GP priors without introducing function noise should be moved to the main text; while those who have previously worked with GPs will be familiar with such issues, this paper is directed towards a wider audience and such clarifications would be helpful for those seeking to replicate the paper’s results. On a related note, I would be keen on learning more about other potential issues with the stability of the optimisation procedure, which does not seem to be discussed upfront in the paper but is key for encouraging the widespread use of such methods.

- The paper contains more than just a handful of silly typos and grammatical errors - too many to list here. This single-handedly detracts from the overall quality of the work, and I highly advise the authors to diligently read through the paper in order to identify all such issues.

 - The references are in an absolute shambles, having inconsistent citation styles, arXiv papers cited instead of conference proceedings, etc. While this is obviously straightforward to set right, I’m nonetheless disappointed that this exercise was not carried out prior to the paper’s submission.

 - The theory presented in Appendix A of the supplementary material appears to be somewhat ‘dumped’ there. Given that this content is crucial for establishing the correctness of the proposed method, linking them more clearly to the main text would improve its readability and give it a greater sense of purpose. I found it hard to follow in its current state.

** Minor **

 - In the introduction there should some mention of deep Gaussian processes which are implicitly a direct competitor to BNNs, and can now also be scaled to millions and billions of observations (Cutajar et al. 2017; Salimbeni et al. 2017). The former is particularly relevant to this work since the architecture can be assimilated to a BNN with special structure for emulating certain kernels.

 - Experiment 5.1.1 is interesting, and the results in Figure 2 are convincing. I would also be interested in seeing how fBNN performs when the prior is misspecified however, which may be induced by using a less appropriate GP kernel. This would complement the already provided insight on using tanh vs ReLU activations.

 - The performance improvement for the experiment on large regression datasets is quite subdued, so it might be interesting to see how both methods compare against each other when deeper BNN architectures are considered.

- With regards to Appendix C.2, which order arccosine kernel is being used here? One can easily draw similarities between the first order arccosine kernel and NN layers with ReLUs, so perhaps it would be useful to specify which order is being used in the experiment.

- Given that the data used for experiments in Appendix C.3 effectively has grid structure, I would be interested in seeing how KISS-GP performs on this task. There should be easily accessible implementations in GPyTorch for testing this out. Given how GPs tend to not work very well on image completion tasks due to smoothness in the kernel, this comparison may also be in fBNNs favour.

- Restating the basic architecture of the BNN being used for the contextual bandits experiment in the paper itself would be helpful in order to avoid having to separately check out Riquieme et al (2018) to find such details.

- I wonder if the authors have already thought about the extendability of their proposal to more complex BNN architectures such as Bayesian ConvNets?


-- Recommendation --

Whereas several ICLR submissions tend heavily towards validation by way of empirical evaluation, I find that the theoretic contributions presented in this paper are by themselves interesting and well-developed, which is very commendable. However, there are multiple telling signs of this being a rushed submission, and I am less inclined to argue ardently for such a paper’s acceptance. Although the paper indeed has its strong points, both in terms of novelty and varied experimental evaluation, in view of this overall lack of finesse and other concerns listed above, I think that the paper is in dire need of a thorough clean-up before being published.

Pros/Cons summary:

+   Interesting concepts that extend beyond empirical fixes.
+   Defining more interpretable priors is a very pertinent topic in the study of BNNs.
+   The presented ideas could potentially have notable impact.
+   Illustrative experiments and benchmark tests are convincing.
-   Not enough connection to MC dropout.
-   Choice of experiments and description of stochastic processes overly similar to other recent widely-publicised papers. It feels on trend, but consequently also somewhat reductive.
-   More than a few typos and grammatical errors.
-   Presentation is quite rough around the edges. The references are in a particularly dire state.

---

> ### Author Response · Authors · 2018-11-22
> **Thank you for your appreciation of our contributions. And we apologize if you had a hard time reading this paper.**
>
>
> Q1, Dropout
> Dropout and its extensions (Gal & Ghahramani, 2016; Gal et al, 2017) are indeed important BNN approaches and have demonstrated success in multiple tasks (Adam D. Cobb, 2018). We have added discussions in both the introduction and related works. In fact we did have direct comparisons with dropout in the contextual bandits problem. And in regression experiments, we compared with stronger baselines (Zhang et al, 2018) than dropout (Gal & Ghahramani, 2016), as shown by their experimental results in the paper.
>
> Q2, New GP methods
> Thanks for your reminder. AutoGP (Krauth et al, 2017) extends Gaussian processes through scalability, kernel and hyper-parameter learning. And it demonstrates enhanced performance over previous approaches. We have cited it in the latest revision.
>
> Q3, Regression with Deeper Networks, Training Time
> Please check our “summary of new revision”, in particular 5 & 6.
> We evaluated the training time for fBNNs and BBB in the large scale regression benchmarks. BBB takes 1,200 seconds for one run and fBNNs take 12,000 seconds for one run, about 10 times longer. Currently we are also working on follow ups to reduce computational costs.
>
> Q4, Injected Noise
> Due to space limitations, we didn’t bring it up to the main paragraph but added a footnote referring to the appendix, which can direct interested readers there. Another potential issue on stability is when we estimate log density derivatives using SSGE (Shi et al, 2018), if samples of q(f) happen to collapse together, the kernel matrix used in SSGE (note elements of this kernel matrix is k(f, f’) instead of k(x, x’)) will also be ill-conditioned. Fortunately, this issue is also addressed by the injected noise approach, by making samples more diverse.
>
> Q5, Typos & References & Proofs
> We feel shame and apologize sincerely for all typos, misused references, and unclear statements. We have fixed all typos and misused references we found. For the proof in Appendix A, we are thinking about better ways to present it.
>
> Minor
> Q1, DGP models
> We have added discussions on DGP models based on your advice. Indeed, (Cutajar et al, 2017) demonstrated close connections between BNNs and DGPs, and proposed weight-space variational inference, which resembles BNNs. (Salimbeni et al, 2017) presented better inference technique for DGPs, which might shed light on BNN research.
>
> Q2, Periodic Extrapolation with mis-specified priors
> We add new results in Figure 2, where we have fBNNs and GP trained using a RBF prior. As shown, fBNNs still fit the training data well, generated reasonable uncertainties, and resembled the GP results. We also made two videos recording the training progresses for the periodic extrapolation experiment, on both RBF and PER+RBF priors. https://drive.google.com/drive/folders/1O_uGu-Drn4TmLalMeQFe1ss2wfkDuW0g?usp=sharing
>
> Q3, Arccosine Kernel
> We used order-1 Arcosine kernel. While close connection exists between order-1 Arccosine kernel and single-layer ReLU networks, several factors still prevent BBB from performing well. Firstly, we used single-hidden-layer network instead of single-layer network, thus the arccosine prior is not the same as BBB prior. Secondly, BBB uses factorized Gaussian for variational inference, which pays large description length cost, hindering it from fitting training data.
>
> Q4, KISS-GP on Grid Structure
> Good question. The texture extrapolation experiment is borrowed from KISS-GP paper (Wilson et al, 2014). KISS-GP can indeed generate good extrapolations if the the data has the grid structure. However, our approach doesn’t assume grid structure, hence it’s more general in the sense that it’s still applicable when the images are not aligned.
>
> Q5, Details on the contextual bandits experiment
> We have added more details for the experiment.
>
> Q6, Bayesian ConvNets
> Our fBNNs is model-agnostic, therefore it should be able to work across different networks, such as CNN and RNN. We indeed used a CNN network in our texture extrapolation experiment (not a Bayesian CNN however). We used a CNN to generate the predictive image starting from a random noise, resembling a GAN generator.
>
> Q7, Choice of experiments and description of stochastic processes
> Firstly, fBNNs perform a series of standard BNNs evaluation experiments, such as x^3, regression and contextual bandits. Moreover, fBNNs perform well in extrapolation tasks including periodic structures, piecewise structures and textures, while none of previous BNNs can do. We think these experiments are strong enough to demonstrate fBNNs' performance. On the other hand, we are also excited to explore the potentials and limitations of fBNNs in extended settings such as model-based RL, which we leave as future work.
>
> For stochastic processes, we referred to Neural Processes for the description of exchangeability and consistency. We think this maintains consistency of notations, which makes it easier to follow for later researchers.

---

> ### Comment · AnonReviewer2 · 2018-11-26
> **Post-rebuttal Update**
>
> I appreciate the addition of the suggested references, but also find their inclusion fairly artificial. In the latest revision, they are simply tacked on to lists of similar references. The presentation has been improved since the orginal submission and is a big step forward towards bringing the paper closer to acceptance. However, I still think that the English writing could be improved further.
>
> Something which I’ve only picked up on in the latest revision is that the large-scale experiments section only features one truly large dataset (GPU; 241600 datapoints). In comparison, the latest state-of-the-art GPs and DGPs which are said to “still suffer for very big dataset[s]“  are evaluated on datasets with millions and even billions of points in their respective papers.
>
> I have read the other reviews and the author rebuttal. Overall I think that although the authors have incorporated many of the changes suggested by the reviewers, these points have mostly been addressed in a cosmetic manner. I would invite the authors to reflect more carefully about the suggestions and rewrite the paper in such a way as to more effectively showcase their contribution in view of these aspects. At this time, I am not inclined to improve my original score.

---

### Official Review · AnonReviewer1 · 2018-11-02
**Interesting paper on functional variational inference but more analysis is needed re the approximations**

**Rating:** 6
**Confidence:** 3

**Review:**

This paper presents a new variational inference algorithm for Bayesian neural network models where the prior is specified functionally (i.e. through a stochastic process) rather than via a prior (e.g.) over weights. The paper is motivated by the maximization of the evidence lower bound defined on stochastic processes, which is itself motivated as the minimization of the KL between the approximate posterior and the true posterior processes.

The paper relies heavily on Th 1, which is proved in the appendix, which states that the KL divergence between two stochastic processes is equivalent to the supremum of the marginal KL divergence over all finite sets of input locations. This yields a GAN-like objective where the ELBO is minimized wrt the input subsets and maximized wrt the approximate posterior. Obviously, as the former minimization is unfeasible, the authors proposed two additional approximations: (1) Restrict the size of the subset to search for; (2) replace the mimimization step with a sampling/average procedure. From the theoretical standpoint, I believe ths is the major defficiency of the paper, as these approximations are not justified and it is not clear, theoretically, how they relate to the original objective. In fact, for example on the case of Gaussian process priors, it looks too good to be true that one can have a KL-divergence over low-dimensional distributions instead of handling N-dimensional (fully coupled) distributions. It is unclear what is lost here (whereas in well-known sparse variational methods such as that of Titsias, one knows how the sparse model relates to the original one).

Only the first experiment compares to a GP model, where it is shown that the solution given by fBNN (which was seeded with the GP solution) is not better (if not slightly worse than the GP’s). As recommended by Matthews et al (2018), all the experiments should compare to a base GP model.

Other Comments:
The paper claims that the method estimates reliable uncertainties. However, there is not an objective evaluation of this claim (as in the predictive posteriors are well-calibrated).
Why aren’t hyper-parameters estimated using the ELBO?
In Figure 2, why are the results so bad for BBB? This is very surprising.
How does the approach relate Variational Implicit Processes (Ma et al, 2018)?
Most of the experiments in the paper assume 1 hidden layer. In the case of deeper architectures, how can one specify a prior over functions that is “meaningful”?
Most (all?) the experiments are specific to regression. Is there any limitation for other likelihood models?
How does the approach compare to inference in implicit models?
In the intro “practical variational BNN approximations can fail to match the predictions of the corresponding GP”. Any reference for this?
I believe the paper should also relate to the work of Matthews et al (2015)


References
(Ma et al, 2018) Variational Implicit Processes
(Matthews et al, 2018) Gaussian process behavior in wide deep neural networks
(Matthews et al, 2015) On Sparse variational methods and the Kullback-Leibler divergence between stochastic processes

---

> ### Author Response · Authors · 2018-11-06
> **Thank you for your detailed reviews. We address your concerns below.**
>
>
> Q1: Approximations incurred by the adversarial and sampling-based methods
> As stressed in the paper, such finite approximations share similar characteristic with GANs (Goodfellow et al, 2014). In GANs, a discriminator classifies between generated samples and real samples. The discriminator computes the JS divergence between the sampling distribution and the true data distribution, **if the discriminator is expressive enough and reaches global optimum**. However, it can never have unlimited capacity. And in practice, the discriminator is only trained several steps in each iteration. Despite these problems, GANs have become the dominant approach to generative modeling in recent years.
>
> Our functional VI has similar characteristic with GANs. Under ideal assumptions that we can search over all finite locations, we are guaranteed to find the exact KL divergence between stochastic processes. In practice, limiting the measurement points size is similar to using a limited-capacity discriminator in GANs. Based on the success of GANs, the approximation here is quite acceptable.
>
> Beyond that, we present a new consistency theorem (Theorem 3) stating that, under mild assumptions, both functional VI algorithms obtain the optimum at the true posterior GPs if measurement points contain all training locations.
>
> Q2: Comparisons with GPs
> Please take a look at our “summary of contributions” and “summary of new revision”. Empirically, we have direct comparisons with GPs in the periodic extrapolation (Section 5.1.1), contextual bandits problem (Section 5.3), bayesian optimization (Appendix C.2), and large scale regression (Appendix C.5).
>
> Furthermore, we stress our contributions more on Bayesian neural networks. Intrinsically, fBNNs lie within scope of BNNs. By introducing functional variational inference, fBNNs demonstrated dominating performances (uncertainty, extrapolation, model-agnosticity, predictive ability) over previous BNNs approaches.
>
> Q3: Evaluation of uncertainties.
> In this work, we demonstrated in various ways that fBNNs generate reliable uncertainties, which is also recognized by AnonReview 2. Qualitatively, our toy experiments demonstrate varying uncertainties near/far from training points. Quantitatively, fBNNs achieved state-of-art on the contextual bandits problem, which was proposed to serve as a standard benchmark to test uncertainties (Riquelme et al, 2018).
>
> Q4, Hyperparameters not estimated by ELBO.
> Jointly optimizing prior hyperparameters and variational parameters has always been a problem in BNNs, including traditional weight space approaches. For instance, before a variational BNN has learned to make reasonable predictions, it may decide to set the observation variance very large and explain all the observations as noise. Our fBNNs suffer from this problem just as much as other variational BNNs, and we leave effective BNN hyperparameter tuning to future work.
>
> Q5, Bad BBB performance, relation to VIP (Ma et al, 2018), deep network priors
>
> VIP (Ma et al, 2018) somewhat resembles an inverse of our method, that they specify a BNN prior over weights, then perform inference using GP techniques. So BBB, VIP both use weight priors. The poor performance of BBB highlights the contribution of fBNNs: we specify priors and perform inference over functional outputs.
>
> We have multi-layer results in Figure 1, Table 3, Table 4 and Table 5. We used 5 hidden layers in Figure 2 and 2 hidden layers in Figure 3.
>
> Q6: Other likelihood models?
> As shown in Eqn 5, fELBO only requires to compute logp(y|f), therefore, it can be used for all likelihoods.
>
> Q7: Inference in implicit models?
> Most implicit models are motivated to use more complicated variational posteriors (Mescheder et al, 2017; Huszár 2017; Shi et al, 2017) or more complicated likelihoods (Tran et al, 2017). In contrast, our method is motivated to solve the BNNs prior problem and borrows the techniques in implicit models to deal with implicit outputs. Indeed, all previously proposed implicit estimation techniques can be adopted in the fBNNs mechanism and better technique will have better performance.
>
> Q8: In the intro “practical variational BNN approximations can fail to match the predictions of the corresponding GP”. Any reference for this?
> Hamiltonian Monte Carlo (HMC) does exact inference asymptotically, thus it performs similarly to the corresponding GP. However, as we shown in Figure 1, BBB under variational inference even fails to match the predictions of HMC. So this demonstrates the argument.
>
> Q9: The work of Matthews et al (2015)
> Thanks. We added the citation and discussed the difference.

---

### Official Review · AnonReviewer3 · 2018-11-06
**This paper concerns the fitting of variational Bayesian Neural Network approximations in functional form and considering matching to a stochastic process prior implicitly via samples.**

**Rating:** 7
**Confidence:** 4

**Review:**

Overall Thoughts:

I found this paper to be very interesting and to address a topic that I think will be of interest to the community. The gap between theoretically advantageous stochastic processes like the GP and the computational efficiency of finite BNNs is a topic not yet fully understood and I believe this paper has some useful points to make. I would be very interested and grateful to hear the authors’ thoughts on the comments/questions below.

Specific Comments/Questions:

I would prefix this discussion with the fact that this is quite a dense (and long) paper on a number of topics so while I hope that I have understood the essence of the approach I apologise if I have missed something and hope the authors will be able to correct me.

I follow the point that it is less clear how finite variational deep BNNs relate to GPs and would agree that finding such an agreement would be a topic of interest. Is the approach taken in the paper not quite close conceptually to the variational sparse GP of Titsias? In that paper, effectively a functional bound is also being taken (i.e. an approximation of a full GP with another GP). So a stochastic process is approximating a full GP. In addition, the approximating GP is defined by a set of samples (the pseudo-input locations) that are optimised as variational parameters. The variational bound is defined in the function domain. There is also alignment with the Stein gradient estimation - the pseudo-input locations are used to define a Nystrom approximation to the full GP kernel. This would seem equivalent to the approximation being performed in (2) as long as the kernel used for the eigenfunctions is the same as that of the full GP.

Following from the above, I would be very interested to directly contrast the differences to the Titsias approach - would it be possible to add it as a baseline to the experiments? In particular, there are potentially differences due to differences between the Stein kernel and the full GP kernel (it might be best to have both kernels the same if they are not already the same in all experiments - sorry, I couldn’t tell). In the variational-GP, the pseudo-input locations are optimised directly under the bound and so whilst they are sensitive to initialisation, the optimisation is stable and guaranteed to converge to a local optimum. It is unclear to me how stable the adversarial problem in Sec3.2 is. In the proposed sampling variant, it isn’t clear to me that it is a safe procedure to follow - taking a random subset from the training data and weights on c seems rather heuristic - are there any guarantees?

There would also seem to be some connections with the recent approaches on (conditional) neural processes - perhaps the authors might like to comment on this?

For a number of the GP priors in the experiments, it might be quite hard for a BNN with ReLu activations to match the posteriors? Would it be worth trying with other (more smooth) activation functions?

Overall the results are interesting - would it be possible to include comparisons to the variational-GP (at least for the small-scale experiments)? It would be interesting to contrast their complexity as well if they get stuck on the large-scale ones. For some of the experiments would it be possible to include histograms rather than just error bars to check that the error distributions are similar?

For the appendix BayesOpt experiment - would it be possible to use Thompson sampling as the acquisition function? To my mind this would evaluate the predictive density more directly since it mitigates the effect of a particular choice of acquisition function on the performance. Also would a comparison with the full GP not be appropriate?

---

> ### Author Response · Authors · 2018-11-22
> **Thanks for your appreciation of our paper. We address your concerns below.**
>
>
> Q1, Comparison with Titsias approach (variational inducing points (VIP))
>
> While both fBNNs and VIP share certain features (sets of input points, Nystrom approximation), these play very different roles in the two methods.
>
> Our measurement points are not analogous to the VIP inducing points. VIP uses the inducing points to define the variational approximation, whereas fBNNs use the stochastic net for this purpose. Our measurement points are used only to estimate the KL term; our number of measurement points is typically much smaller than the number of inducing points for VIP.
>
> The Nystrom approximation we use in SSGE is not analogous to the one used in VIP, as the two approximations are computed in different spaces. In sparse GPs, the kernel matrix is computed in the input space, i.e., k(x, x’), while in our SSGE gradient estimator it’s computed in the space of vectors of function values: k(f, f’), f, f’ ~ q(f). The only purpose of this kernel matrix is to approximate derivatives of \log q(f).
>
> Some other differences from VIP: firstly, BNNs provide explicit function samples (see Figure 3) while GPs only predict marginals. Secondly, our fBNNs can use arbitrary-size network to increase expressiveness, while VFE is heavily limited by the inducing points. In Table 4, we demonstrated fBNNs closed the gap between BBB and VFE for small datasets. In Table 5, we turned to SVGP, the mini-batch version of VFE, and demonstrated fBNNs’ superiority over SVGP for large datasets. We found the inducing points strongly hampered SVGP’s performance.
>
> Q2, Adversarial Functional VI and Sampling-Based Functional VI
> Throughout our experiments we mainly investigated sampling-based functional VI, as we found that sampling-based functional VI was very stable. As stressed in the paper, sampling measurement points from a distribution c is equivalent to attaching varying importance to different regions. Because the measurement points are random in each iteration, sampling-based functional VI will force the variational posterior to perform well across the domain.
>
> Beyond that, we present a new consistency theorem (Theorem 3) stating that, under mild assumptions, both functional VI algorithms converge to the true GP posterior if measurement points contain all training locations.
>
> Q3, Comparison with NP
> (Conditional) Neural processes (NP) try to learn any posterior distribution conditioned on any data points. Specifically, they derived the connection between posteriors with different conditioning points, and performed further approximation to get a tractable objective.
>
> Our approach is fundamentally different with NP in the sense of learning THE posterior given particular data points, which enables more efficient learning. In concrete, NP encounters difficulties in three situations,
>     High Dimension. The posterior distributions in high dimensional space are more complicated, thus it is much more difficult to model posteriors conditioned on any data points. Besides, NP samples a function from prior in each iteration, which is not sample-efficient enough to model high dimensional distributions. (Note they used at most 2 dimensional input in the paper.)
>     Large Scale. NP cannot scale to large scale datasets in two senses. Firstly, its architecture cannot even afford to condition on the whole dataset because of memory issues. Secondly, NP is trained on small sets of points, which might be hard to generalize to large datasets.
>     Unknown prior. NP relies heavily on the prior, thus it learns nothing if the prior is not known. However, in most cases we are only provided a single dataset without the functional prior. In contrast, fBNNs still generate reasonable predictions even with a mis-specified prior (see Figure 2).
>
> However, we think NP is still a nice approach despite these problems, .
>
> Q4, Activations
> fBNNs are applicable to all activations. It is true that proper activation function will boost the performance. For the implicit prior problem, we used TanH as activation. For most of the experiments, we used ReLU with factorized Gaussian distributions. Although they worked well surprisingly, it is still important to work on better variational posterior structures, including activation, weight distribution, network and etc.
>
> Q5, Bayesian Optimization
> In MES (Wang and Jegelka, 2017), to get samples for minimizer y^{\star}, we have to use random feature instead of a full GP. Actually, full GP algorithms like PI and EI performs worse than MES as shown in (Wang and Jegelka, 2017). And MES indeed used Thompson sampling to get samples for minimizer y^{\star}. For evaluating the predictive uncertainty using Thompson sampling, please refer to our contextual bandits experiment, where we used Thompson sampling and achieved state-of-art performance.

---

### Author Response · Authors · 2018-11-06
**Summary of new revision**


1, We add theoretical results on adversarial and sampling-based functional VI in Theorem 3.
2. We add the KL divergences between two conditional stochastic processes in Appendix B.1.
3, We compare function-space inference with weight-space inference theoretically in Appendix B.2, where the weight prior is equivalent to the functional prior.
4, In Figure 2, we add new experimental results that fBNNs were trained using a RBF kernel. The results manifest that fBNNs still generated reasonable predictions even with a mis-specified prior. We made two videos recording the training progress for the periodic extrapolation experiment, on both RBF and PER+RBF priors. https://drive.google.com/drive/folders/1O_uGu-Drn4TmLalMeQFe1ss2wfkDuW0g?usp=sharing
5, In Appendix C.4, we add regression results with varying network sizes and compare with VFE (Titsias, et al, 2009). We found BBB totally failed with 5 hidden layers of 500 units, which is in accord with our Figure 1 observation. In contrast, fBNNs worked well consistently and closed the gap between BBB and VFE.
6, In Appendix C.5, we add large scale regression results with 5 hidden layers of 100 units. We compared BBB, fBNNs and SVGP (Hensman et al, 2014). BBB and fBNNs demonstrated obviously better performance over Table 2, where we used 1 hidden layer of 100 units. SVGP has its performance dropping with dataset size increasing.
7, We corrected typos and misused references we found. We added references in introduction and related works based on the reviewers’ suggestions.

[References]
(Hensman et al, 2014) Scalable Variational Gaussian Process Classification
(Goodfellow et al, 2014), Generative Adversarial Nets
(Neal, 2012) MCMC using Hamiltonian dynamics
(Ma et al, 2018) Variational Implicit Processes
(Matthews et al, 2015) On Sparse variational methods and the Kullback-Leibler divergence between stochastic processes
(Riquelme et al, 2018) Deep Bayesian Bandits Showdown
(Huszár, 2017) Variational inference using implicit distributions
(Mescheder et al, 2017) Adversarial variational bayes: Unifying variational autoencoders and generative adversarial networks
(Shi et al, 2017) Kernel implicit variational inference
(Tran et al, 2017) Hierarchical implicit models and likelihood-free variational inference
(Zhang et al, 2018) Noisy Natural Gradient as Variational Inference
(Bui et al, 2016) Deep Gaussian Processes for Regression using Approximate Expectation Propagation
(Gal & Ghahramani, 2016) Dropout as a bayesian approximation: Representing model uncertainty in deep learning.
(Gal et al, 2017) Concrete dropout.
(Adam D. Cobb et al, 2018) Loss-Calibrated Approximate Inference in Bayesian Neural Networks
(Krauth et al, 2017) AutoGP: Exploring the Capabilities and Limitations of Gaussian Process Models
(Cutajar et al, 2017) Random Feature Expansions for Deep Gaussian Processes
(Titsias, et al, 2009)Variational Learning of Inducing Variables in Sparse Gaussian Processes
(Shepherd et al, 2017) Mapping Gaussian Process Priors to Bayesian Neural Networks
(Li and Turner, 2018) Gradient Estimators for Implicit Models
(Salimbeni et al, 2017) Doubly Stochastic Variational Inference for Deep Gaussian Processes
(Wilson et al, 2014) Fast Kernel Learning for Multidimensional Pattern Extrapolation
(Rasmussen and Williams, 2006) Gaussian Processes for Machine Learning
(Wang and Jegelka, 2017) Max-value Entropy Search for Efficient Bayesian Optimization

---

### Author Response · Authors · 2018-11-06
**Summary of Contributions**

1. We present the theorem to compute KL divergence between stochastic processes. Stochastic processes have various implications in machine learning research, including Gaussian Processes, Bayesian neural networks and so on. Furthermore, generative models including VAEs (Kingma et al, 2013) and GANs (Goodfellow et al, 2014) can also be viewed as the discretization of generated functions (of stochastic processes). Therefore, our presented theorem lays the foundation and opens up new views for many research areas.

2. With KL divergence of stochastic processes, we present two practical functional variational inference methods: the adversarial functional VI, which is closely related to the popular GAN-like training procedures; the sampling-based functional VI, which provides an uniformly maximization objective and avoids the instability of adversarial training. Our functional variational inference methods are quite general and ready to be adopted in many scenarios, including BNNs, VAEs and etc.

3. We pin out the mis-specified weight-space prior problem in BNNs. We show that BNNs can perform unpredictably when varying network sizes. Alternatively, we propose fBNNs to perform functional VI over the outputs of a BNN. To the best of our knowledge, we are the first to apply functional priors and perform functional VI over BNNs.

4. Empirically, we conduct multiple experiments to demonstrate our fBNNs’ superiority over standard BNNs
	1) Reliable uncertainties (see contextual bandits, Bayesian optimization and Figure 1).
	2) Sensible extrapolations (see extrapolating periodic structures, piecewise structures and grid texture patterns).
	3) Predictive abilities (see regression experiments).
	4) Scalability (see large scale regression).
	5) Model-agnostic (see the texture pattern experiment where we used CNNs).

5. Comparisons to GPs
	1) fBNNs enable fast inferences compared to GPs.
	2) fBNNs enable richer prior structures, including explicit priors such as student-t process, and implicit priors such as piecewise priors.
	3) fBNNs enable sampling coherent functions (at global level) since the weights of fBNNs can be viewed as global latent variables.


However, as a pioneering work, we don’t expect fBNNs to solve all problems, and we don’t expect fBNNs to totally outperform GPs in all aspects, which has been more than 20 years’ fruitful research. However, we pioneered a new promising research area that connects closely between deep learning networks and statistical theories. Overall, our fBNNs push forward the frontier of BNNs research. We show that it is more desirable to work on the functional-space in view of stochastic processes than working on the weight-space. It also provides new techniques to the research of more general stochastic processes.

---

### Meta-Review · Area_Chair1 · 2018-12-13
**Good work which can become more mature with further experiments**

**Confidence:** 3
**Recommendation:** Accept (Poster)

**Metareview:**

This paper shows a promising new variational objective for Bayesian neural networks. The new objective is obtained by effectively considering a functional prior on the parameters. The paper is well-motivated and the mathematics are supported by theoretical justifications.

There has been some discussion regarding the experimental section. On one hand, it contains several real and synthetic data which show the good performance of the proposed method. On the other hand, the reviewers requested deeper comparisons with state-of-the art (deep) GP models and more general problem settings. The AC decided that the paper can be accepted with the experiments contained in the new revision, although the authors would be strongly encouraged to address the reviewers’ comments in a “non-cosmetic manner (as R2 put it).